# Proteomic Analysis of Hydromethylthionine in the Line 66 Model of Frontotemporal Dementia Demonstrates Actions on Tau-Dependent and Tau-Independent Networks

**DOI:** 10.3390/cells10082162

**Published:** 2021-08-22

**Authors:** Karima Schwab, Valeria Melis, Charles R. Harrington, Claude M. Wischik, Mandy Magbagbeolu, Franz Theuring, Gernot Riedel

**Affiliations:** 1School of Medicine, Medical Sciences and Nutrition, University of Aberdeen, Foresterhill, Aberdeen AB25 2ZD, UK; karima.schwab@abdn.ac.uk (K.S.); v.melis@abdn.ac.uk (V.M.); c.harrington@abdn.ac.uk (C.R.H.); cmw@taurx.com (C.M.W.); 2Charité—Universitätsmedizin Berlin, Hessische Str. 3-4, 10115 Berlin, Germany; mandy.magbagbeolu@charite.de (M.M.); Franz.Theuring@charite.de (F.T.); 3TauRx Therapeutics Ltd., 395 King Street, Aberdeen AB24 5RP, UK

**Keywords:** mouse model, tau protein, frontotemporal dementia, proteomics, hydromethylthionine, LMTM

## Abstract

Abnormal aggregation of tau is the pathological hallmark of tauopathies including frontotemporal dementia (FTD). We have generated tau-transgenic mice that express the aggregation-prone P301S human tau (line 66). These mice present with early-onset, high tau load in brain and FTD-like behavioural deficiencies. Several of these behavioural phenotypes and tau pathology are reversed by treatment with hydromethylthionine but key pathways underlying these corrections remain elusive. In two proteomic experiments, line 66 mice were compared with wild-type mice and then vehicle and hydromethylthionine treatments of line 66 mice were compared. The brain proteome was investigated using two-dimensional electrophoresis and mass spectrometry to identify protein networks and pathways that were altered due to tau overexpression or modified by hydromethylthionine treatment. Overexpression of mutant tau induced metabolic/mitochondrial dysfunction, changes in synaptic transmission and in stress responses, and these functions were recovered by hydromethylthionine. Other pathways, such as NRF2, oxidative phosphorylation and protein ubiquitination were activated by hydromethylthionine, presumably independent of its function as a tau aggregation inhibitor. Our results suggest that hydromethylthionine recovers cellular activity in both a tau-dependent and a tau-independent fashion that could lead to a wide-spread improvement of homeostatic function in the FTD brain.

## 1. Introduction

Tauopathies, such as Alzheimer’s disease (AD) and frontotemporal dementia (FTD), are distinct neurodegenerative disorders with overlapping pathology. A common feature is the deposition of abnormal tau aggregates [1,2,3] which, under physiological conditions promotes microtubule assembly, stabilises axons and allows axonal branching and axonal transport [4]. For FTD, cases are characterised by their pathology; most cases presenting with either tau or TDP-43 pathology [5]. In the diseased brain, small oligomeric aggregates of tau are formed initially, with subsequent accumulation of tau in paired helical filaments (PHFs) and these changes contribute to synaptic dysfunction, microtubule collapse and neuronal death [6]. While several mutations in the longest human CNS tau isoform (htau40) appear to promote tau aggregation and have been associated with early onset frontotemporal dementia with parkinsonism linked to chromosome 17 (FTDP-17) [7,8,9,10], relatively little is known about how these mutations affect the global proteome. This work constitutes one of the first attempts to determine such global alterations in brain protein expression as a result of drug treatment.

Clinical proteomics have been used for almost three decades to discover diagnostic markers in neurodegenerative disorders (see [11,12,13,14,15,16,17,18,19,20]). These studies concentrated on AD and have used brain tissue, cerebrospinal fluid and blood as substrates for analysis. Low reproducibility was found between independent studies possibly due to technical variability, sample treatment prior to proteomic analysis and because of heterogeneity of the patient cohort as a result of failure to separate out their co-morbidities. The more recently applied endophenotype based data correlation approach appears to be more reliable [16]. However, limited proteomic studies are available for FTD (for review see [21,22,23,24,25]). For example, Hu and co-workers identified 151 differentially regulated proteins for FTD, but these only yielded 78% specificity for FTD with TDP-43 pathology (FTD-TDP) over FTD with tau pathology (FTD-tau) [26]. By contrast, Teunissen et al. reported 56 differentially regulated proteins, which is less than 4% of the 1914 identified proteins, and only 10 of these differed between those FTD cohorts characterised by TDP-43 and tau pathologies [27]. Intriguingly, YKL-40 (a member of the glycosyl hydrolase 18 family with a role in inflammation) was increased in both forms of FTD, while catalase was reduced, when compared to controls and patients with AD, dementia with Lewy bodies or vascular dementia. Comparisons of CSF from AD or FTD patients with control subjects yielded only few proteins that altered abundance specifically with disease status. A multi-centre surface-enhanced laser desorption/ionisation time-of-flight mass spectrometry (SELDI-TOF) study revealed 15 putative biomarkers in CSF of AD and FTD patients that were differentially regulated [28]. Seven of these identified proteins (including cystatin C and chromogranin A peptide) also differentiated between the two dementia disorders. However, the proteins identified did not match those of previous studies using CSF samples [29,30], a finding which was ascribed to their lower solubility and abundance in CSF compared to brain [31].

The majority of studies on brain tissue for FTD have focussed on cases with TDP-43 pathology and have examined extracts derived from prefrontal cortex or hippocampus with more than 50 proteins changing abundance in the diseased brains (see [24] for a summary). Using laser capture microdissection, high resolution LC-MS/MS and labelling-free quantitation methods, Gozal and colleagues confirmed 54 proteins selectively increased, while 19 proteins out of a total of 1252 selectively decreased abundance in dentate gyrus granule cells of patients with the ubiquitinated type of FTD [32]. Similarly, earlier work by Schweitzer and colleagues on frontal cortex from FTD patients had returned 48 proteins with matrix-assisted laser desorption/ionisation (MALDI) TOF that were of different abundance in FTD patients relative to controls and functionally classified into cytoskeleton, metabolism, oxidative stress, proteolysis, signal transduction and others [33]. The proteolytic enzyme ubiquitin C-terminal hydrolase L1 was included in this list and was particularly reduced in a patient expressing the P301L mutation of FTDP-17 and may explain how the mutation disrupts interactions of the C-terminal domain of tau with the proteasome [34].

In experimental AD models, overexpression of human wild-type tau under the control of the neuronal enolase promotor followed by 2-DE of cortical tissue at 6 and 12 months yielded a total of 19 proteins changing in abundance [35]. These included secernin 1, V-type proton ATPase subunit E1, glyceraldehyde 3-phosphate dehydrogenase, dihydropyrimidinase-related protein 2, proteasome subunit alpha type-4, and NADH dehydrogenase [ubiquinone] iron-sulphur protein 8. These proteins were increased at 6, but decreased at 12 months, and for which neuroprotective roles were suggested.

For 3xTg AD mice, which overexpress tau carrying the P301L mutation and ß-amyloid precursor protein carrying the Swedish double mutation, hippocampal tissue was studied at 12 months of age with a focus on *O*-GlcNAcylated proteins. In this study, 14 differentially expressed proteins that were *O*-GlcNAcylated were involved in glucose homeostasis, energy metabolism, cytoskeletal network and neurotransmission [36]. By contrast, oxidative phosphorylation was the focus of another study using hippocampal tissue from 3xTg AD mice. Sixty-four mitochondrial and nuclear proteins were differentially expressed [37]. However, a clear link with tau expression remains elusive given the potential interaction between amyloid and tau in these mice. A more appropriate FTD-like model therefore is established through overexpression of P301S/L tau only. In a two-dimensional difference gel electrophoresis (2D-DIGE) approach, spinal cord from 7-month-old mice was homogenised and returned a total of 32 differentially regulated proteins; four of these proteins, heat shock protein 27 (Hsp27), peroxiredoxin-6, apolipoprotein E, and latexin, were all enhanced in astrocytes suggesting a neuroprotective function [38]. Similarly, heat shock proteins were amongst the upregulated proteins in Tg4510 mice which also expressing human tau carrying the P301S mutation [39]. These data strongly indicate the need for a more comprehensive study of the brains of FTD models. We selected the line 66 (L66) mouse and compared whole brain tissue with age-matched wild-type mice using two-dimensional electrophoresis (2-DE) followed by nano liquid chromatography tandem mass spectrometry (nano-LC-ESI-MS/MS). L66 has been extensively characterised behaviourally [40], with respect to tau expression and localisation [41], in terms of cholinergic status and inflammation [42] and in synaptic release of glutamate [43]. L66 mice overexpress the largest human 2N4R isoform (441 amino acid residues) carrying aggregation-promoting mutations P301S and G335D [40,44].

Prevention of tau aggregation is a valid therapeutic strategy since numerous studies have confirmed a correlation between the degree of tau aggregation and the severity of dementia [12,13,37,45]. The most promising approach, targeting the inhibition of tau aggregation and its clearance, is based on methylthioninium chloride (MTC) and its stably reduced form *N,N,N′,N′*-tetramethyl-10*H*-phenothiazine-3,7-diaminium bis (hydromethanesulfonate) (leucomethylthioninium bis(hydromethanesulfonate; LMTM). LMT has recently been assigned the International Nonproprietary Name “hydromethylthionine”, recognising it as chemically and pharmacologically distinct from MTC. The methylthioninium (MT) moiety can exist in oxidised (MT^+^) and reduced (LMT) forms and we have reported recently that LMTM blocks tau aggregation and propagation in vitro [46], reverses behavioural deficits and tau pathology in mice [47] and hydromethylthionine has pharmacological activity on brain structure and function in both AD [48] and FTD [49]. In addition to its effect as a tau aggregation inhibitor, MTC has several beneficial effects on pathways relevant to neurodegenerative disorders [50]. It induces mitochondrial biogenesis [51], mitochondrial complex I-IV activity [52,53,54,55,56], activates NRF2-mediated antioxidant response [57], inhibits microglial activation [58] and facilitates tau clearance by promoting the ubiquitin–proteasome-system and autophagy [59,60]. These data confirm that MT acts on multiple intracellular systems. Understanding the pathways activated in L66 by LMTM would therefore instruct on the potential treatment benefit and aid in the development of novel therapeutic strategies. In this study, we report proteomic analysis of L66 mice treated with two dosing regimes of LMTM, an approach that has not been available in other experimental models or in humans.

## 2. Materials and Methods

### 2.1. Animals

Transgenic mice are described in detail elsewhere [40]. Briefly, L66 mice overexpresses the largest human CNS tau isoform (htau40 with 441 amino acid residues), with the mutations P301S and G335D, under the control of the mouse *Thy1*-promotor. These mice express early onset, high tau load in the brain globally and their genetics, behavioural and histopathological phenotypes are reminiscent of frontotemporal dementia. A detailed characterisation of these transgenic mice was reported earlier and male and female mice show similar behavioural and pathological phenotypes [40,41,42].

Female homozygous transgenic L66 and wild-type NMRI litters were bred commercially in isolators (Harlan (now Envigo), Hillcrest, UK) and delivered by truck at least 10 days before experimental work commenced. They were housed genotype specifically in small colonies up to five animals in open housing (Type III, 382 mm × 220 mm) with corn cob bedding and paper strips and cardboard tubes as enrichment (cleaning rota once per week). Holding rooms were on constant temperature (20–22 °C), humidity (60–65%), and air exchange rate (17–20 changes/h) with 12 h light/dark cycle (lights on at 6 am, simulated dawn). Animals had free access to food and water. All animal experiments were performed in accordance with the European Communities Council Directive (63/2010/EU) and a project licence with local ethical approval under the UK Animals (Scientific Procedures) Act (1986) and comply with the ARRIVE guidelines 2.0 [61]. Brains were sent to Berlin (Charité) via courier on dry ice.

For proteomics and protein-based studies, different female mouse cohorts were used for two separate experiments: Experiment 1: untreated wild-type NMRI mice and L66 mice (8–12 per group; Figure 1A); Experiment 2): vehicle and LMTM-treated L66 mice (11–14 per group; Figure 1B). A third independent cohort (9–13 mice per group, including NMRI and L66, treated with vehicle/LMTM) was used for histopathology examinations. Animals were allocated to experimental groups based on their starting body weight with equal average body weight for each group.

### 2.2. Drugs and Treatment Cohorts

*N,N,N’,N’*-Tetramethyl-10*H*-phenothiazine-3,7-diaminium bis(methanesulfonate) (LMTM) was supplied by TauRx Therapeutics Ltd., Aberdeen UK. The doses of LMTM are expressed in terms of free methylthioninium (MT) base per animal body weight (mg MT/kg). For vehicle and LMTM treatments, 4-months old mice were assigned to treatment cohorts and dosed with vehicle or LMTM for 8 weeks (seven days per week) and sacrificed at 6 months of age. Compounds were administered in the morning via oral gavage, at a volume of 5 mL/kg body weight. LMTM was dissolved in vehicle (argon-sparged deionised water) and administered within 20 min of dissolution. Treatment regime and dose selection were based on successful lowering of tau pathology and on behavioural phenotype rescue in tau-transgenic mice [47]. Only L66 tau mice were exposed to treatment with LMTM to focus on the correction/alteration of disease-relevant proteins.

### 2.3. Sample Collection

At the age of 6 months, we administered the last vehicle/drug and sacrificed mice by cervical dislocation one hour later. Brains (without olfactory bulbs) were removed rapidly, washed in cold 30 mM 2-ethane sulphonic acid/60 mM sodium fluoride buffer and either snap frozen in liquid nitrogen for proteomics and immunoblotting analyses (two groups of 7–10 mice per group for genotype related effects and three groups of 11–14 mice per group for LMTM related effects) or fixed for 24 h in formalin, embedded in paraffin and sectioned on a rotary microtome (Microm HM325, Leica Biosystems, Nussloch, Germany) at 5µm for histopathology (four groups of 9–13 mice per group).

### 2.4. Immunohistochemistry and Neuronal Cell Counting

Coronal brain sections, 5 µm thick and mounted onto glass slides (three sections per slide), were deparaffinised, boiled for antigen retrieval in 10mM citric buffer and then incubated in 0.3% (*v*/*v*) hydrogen peroxidase solution. Sections were blocked for 20 min in blocking solution (0.1% (*w*/*v*) BSA in PBS), incubated in primary antibody (NeuN Clone A60, Merck Millipore, Burlington, MA, USA, diluted 1:11,000), followed by incubation in biotinylated goat anti-mouse IgG secondary antibody (Dako Denmark, Glostrup, Denmark). Primary and secondary antibodies were diluted in blocking solution and reactions were conducted for 1 h at room temperature (RT). Sections were then developed with diaminobenzidine solution (Dako Denmark) and embedded in Neo-Mount (Merck Millipore, USA). Images were taken using a light microscope at a 200× magnification (Carl Zeiss, Jena, Germany) and quantification was conducted by an investigator, blinded with respect to the genotype and the treatment. Neurons expressing NeuN were counted manually in a fixed area in hippocampal CA1, CA3 and dentate gyrus, all at Bregma level −3.16 ± 0.36, as well as in primary motor cortex (MC) at Bregma level 0.74 ± 0.36 [62].

### 2.5. Urea Protein Extraction

For 2-DE, crushed frozen tissue from whole brain (including cerebellum and excluding olfactory bulb) was incubated for 45 min in six volumes of extraction buffer (7 M urea, 2 M thiourea, 2% ampholyte 2-4, 70 mM DTT, 25 mM Tris/HCl pH 8.0, 50 mM KCl, 3 mM EDTA, 2.9 mM benzamidine and 2.1 µM leupeptin) and centrifuged for 45 min at 16,000× *g* and RT. The urea extract (supernatant) was transferred to new tubes and the protein concentration determined with Bradford reagent (Carl Roth, Karlsruhe, Germany) according to standard procedures.

### 2.6. 2-DE and Comparative Gel Analysis

Large scale 2-DE (23 cm × 30 cm × 0.75 mm) and image analyses were conducted as described before [63]. Protein extracts (100 µg) were loaded at the anodic side of the isoelectric focusing (IEF) gel and focused in a gradient between pH 2 and 11 at 8500 Vh (1 h at 100 V, 1 h at 200 V, 16.5 h at 400 V, 1 h at 600 V, 30 min at 1000 V, 10 min at 1500 V, 10 min at 2000 V). After equilibration in Tris-SDS-DTT solution (50mM Tris pH 6.8, 3% (*w*/*v*) SDS and 1% DTT (*w*/*v*)), IEF gels were placed on top of a 15% glycine-SDS second dimension gel and separated at 4 °C in Tris-glycine buffer (25mM Tris and 192 mM glycine; pH 6.8). Gels were fixed overnight in 50% (*v*/*v*) ethanol and 10% (*v*/*v*) acetic acid and silver stained [64]. Gel images with 150 dpi resolution were analysed with the Proteomweaver software version 3.0.9.9 (BioRad, Feldkirchen, Germany) according to the manufacturer’s instructions. The software includes an integrated statistics tool to automatically normalise the data set and calculate average spot intensities, regulation factors (=spot intensity ratio) and *p*-values. Briefly, spots were automatically detected and spot intensities were then normalised to total intensity of all spots (on one gel) and matching was carried out first inside groups and then between groups. Inaccurate spot identification was corrected manually. Spots were considered to be regulated when the relative spot density was significantly different (*p* < 0.05) and if the spot was present in at least 80% of gels in each study group. No further cut-off or post-hoc analyses were applied to the data and all significantly regulated spots with the above-described behaviour were subjected to MS analysis. For the genotype analysis (experiment 1), wild-type NMRI mice were set as reference group, while for LMTM-associated effects (experiment 2), vehicle-treated L66 mice served as reference.

### 2.7. Trypsin Digestion and MS-MS Analyses for Protein Identification

Proteins (120 µg) were separated as stated above by large scale preparative 2-DE gels (23 cm × 30 cm × 1.5 mm) and gels were thereafter stained with MS-compatible silver [65]. All significantly different protein spots were excised from preparative gels and digested using 200 ng trypsin. Digestion was conducted overnight at 37 °C and the enzymatic reaction was terminated the next day with formic acid. Tryptic peptides were analysed by nanoLC-ESI-MS/MS. The MS system consisted of an Agilent 1100 nanoLC system (Agilent, Waldbronn, Germany), PicoTip electrospray emitter (New Objective, Littleton, MA, USA) and an Orbitrap XL mass spectrometer (Bruker, Bremen, Germany).

After trapping and desalting the peptides on an enrichment column (Zorbax SB C18, 0.3 mm × 5 mm, Agilent, Germany) using 1% (*v*/*v*) acetonitrile and 0.5% (*v*/*v*) formic acid for five minutes, peptides were separated on a Zorbax 300-SB-C18 column (75 µm × 150 mm, Agilent, Germany) within 40 min, using a formic acid (0.1%) acetonitrile (5–40%) gradient. The precursor ion was sent into gas filled collision cell for high energy collisional dissociation. The fragment ions were sent to C-trap and transferred to orbitrap for analysis Raw spectra files were generated by Xcalibur 2.1 (Thermo Fisher Scientific, Waltham, MA, USA), uploaded to the MaxQuant platform (version 1.5.1.2, Max Planck Institute of Biochemistry, Martinsried, Germany) and searched against a UniProtKB mouse protein data base (downloaded from www.uniprot.org in December 2015). The MaxQuant software allows protein identification and label-free quantification (LFQ; based on peak intensities of spectra). The search settings were orbitrap as instrument, trypsin as enzyme and two missed cleavages were allowed, Carbamidomethyl (C) was set as fixed modification and Acetyl (protein N-term), Oxidation (M) and Deamidation (NQ) as variable modification. The peptide tolerance was set to 5ppm, the MS/MS match tolerance to 0.5 Da, the peptide spectral matches false discovery rate (FDR) and the protein FDR to 0.01. Further, the site decoy fraction was set to 0.01, the minimal peptide length to 7, the minimal number of matched peptides was 1 and the minimal score for peptides was 40.

In a case where the LFQ intensity of the first hit being >70% then only the first hit is reported. In a case where the LFQ intensity of the first hit is <70% and the LFQ intensity of the second hit >10% then the second hit is reported as well. For raw 2-DE and MS results, see Appendix A; for annotated peptide spectra for protein identification based on one single peptide, see Appendix A.

### 2.8. Generation of Expression Heatmaps, Protein Classification and Pathway Analyses

All identified proteins were clustered according to their expression pattern using the heatmapper online tool (www.heatmapper.ca, accessed on 30 March 2020) with average linkage clustering and the Euclidean distance measurement method. For the generation of these expression heatmaps, numerical information uploaded are transferred to z-scores (number of standard deviation) and the average over all groups was build (NMRI and L66 for Experiment 1; L66 vehicle, L66 15 mg and L66 45 mg for Experiment 2). The z-scores were then assigned a colour depending on their relative location to the population’s average: the average is shown in black; values below the average are displayed red, while values above the average are displayed green. The colour intensity indicates the distance to the population’s average (dark colours being closer to the average than light colours). Further, we classified these proteins according to biological processes using the Panther classification system database (www.pantherdb.org, accessed on 30 March 2020). For pathway analyses, the Ingenuity Pathway Analysis tool (IPA, version 01.04, Ingenuity Systems (now Qiagen Silicon Valley), Redwood City, CA, USA) was used and the search limited to networks in nervous system of mammals with known relationships between proteins. Networks were automatically assigned based on the probability of protein matches and contrasted against chance. On the basis of altered pathways the software predicts putative upstream regulators.

### 2.9. SDS-Page and Immunoblotting

The urea protocol from above was used to extract proteins for immunoblotting replacing the ampholytes with distilled water. Twenty µg protein per lane were separated by SDS-PAGE using stain-free gradient gels (Bio-Rad, Germany) in Tris-glycine-buffer (pH 8.3) containing 192 mM glycine, 25 mM Tris and 0.9% (*w*/*v*) SDS. Proteins were then transferred for 30 min at 5V to a PVDF membrane by semi-dry blotting in Towbin buffer pH 8.3 (200 mM glycine, 25 mM Tris, 0.1% (*w*/*v*) SDS and 20% (*v*/*v*) methanol). Thereafter, membranes were blocked at RT for 1 h in blocking solution (4% (*w*/*v*) BSA in TBS with 0.2% (*v*/*v*) Tween-20), incubated overnight at 4 °C in primary antibody, washed 3 times in TBS-T and incubated for 1 h (RT) in appropriate secondary antibody conjugated to HRP (Dako, Denmark; goat anti-rabbit IgG, goat anti-mouse IgG and rabbit anti-goat IgG diluted 1:5000 in blocking solution). After washing 3 times in TBS-T, membranes were overlaid with ECL solution (GE Healthcare, Chicago, IL, USA) and chemiluminescent signals were detected with the ChemiDoc™ MP imager (Bio-Rad, Feldkirchen, Germany). Primary antibodies used were: AKT (#9272; 1:5000), p-Ser473 AKT (#4058; 1:1000), Erk1/2 (#4695; 1:1000), pThr202/Tyr204 Erk1/2 (#9101; 1:1000), CaMKII (#3362; 1:1000), pThr286 CaMKII (#12716; 1:1000), Synapsin-2 (#85852; 1:10,000), α-synuclein (#4179; 1:2000), TrkB (#4603; 1:1000), P70S6K (#9202; 1:1000), pP70S6K (#9206; 1:5000), 4E-BP1 (#9452; 1:1000), p4E-BP1 (#9459; 1:1000), GSK-3α (#4337; 1:1000), p GSK-3α (#9316; 1:1000), GSK-3β (#12456; 1:1000) and p GSK-3β (#5558; 1:1000) all from Cell Signaling Technology Europe, Frankfurt am Main, Germany. PSD95 (ab12093; 1:1000) and syntaxin (ab112198; 1:5000) were purchased from Abcam (Cambridge, UK). Synapsin-1 (#106001; 1:5000), synaptophysin-1 (#101002; 5:1000) and VAMP2 (#104202; 5:1000) were from Synaptic Systems, Goettingen, Germany. SNAP-25 (#805001; 1:1000) was purchased from BioLegend, San Diego, CA, USA. BDNF (#MABN79; 1:500) was from Merck Millipore (USA). All primary antibodies were diluted in blocking solution. Densitometric quantification was performed using the Image Lab software version 5.0 (Bio-Rad) and data were normalised to stain free total protein loading. Blots for raw data are shown in the Appendix A.

### 2.10. Data Analyses

Proteomics data were analysed by using integrated statistics tools of the Proteomweaver, The MaxQuant and the IPA software following the parameters indicated above and no further post-hoc analyses were applied to the data. Immunoblotting and immunohistochemistry data are expressed as group mean and standard error (S.E.) and statistical analysis was conducted using analysis of variance (ANOVA) with the Bonferroni multiple comparison correction post-hoc test or using the unpaired t-test if appropriate (GraphPad Prism software version 6.00; GraphPad Software Inc., San Diego, CA, USA) and differences were considered significant at *p* < 0.05.

## 3. Results

An LMTM regime similar to the one used in the current study was reported earlier to significantly lower tau pathology in multiple brain regions and to correct behavioural abnormalities seen in L66 mice [47]. We have now in an independent study treated L66 mice with a similar LMTM regime and conducted a hypothesis-free proteomics study to identify key pathways underlying these corrections.

Our large scale 2-DE protocol detected an excess of 2000 spots on average per mouse (Figure 2 and Figure 3). A representative 2-DE protein pattern for L66 against wild-type in experiment 1 (Figure 2 shows an example gel image for a wild-type mouse) revealed 71 spots (=123 protein species; see Figure 4A) that were differentially abundant between genotypes (*p* < 0.05). In contrast, due to the administration of LMTM to L66 (Figure 3 shows an example gel image of a L66 vehicle-treated mouse), 433 spots (=657 protein species, see Figure 4A) were differentially abundant in experiment 2. Seventeen spots (=27 protein species) were common to both analyses (see Venn diagram, Figure 4A). The total of 487 spots (71 + 433 − 17 common to both groups) represent 753 regulated protein species [66], as confirmed by MS. Appendix A shows in detail which spots correspond to which protein species, taking into consideration the LFQ inclusion criteria stated in the Methods section.

Based on the Panther classification data base, altered proteins were organised into biological processes. Of the proteins dysregulated in L66 (vs. wild-type), 28% were metabolic, 12% related to stress response and around 28% exerted cell structure/microtubule stabilisation and neuronal development/neurotransmission functions (Appendix A). Almost 27% of all proteins changing due to LMTM exerted metabolic functions, 6% were underlying the stress response, and more than 40% are involved in various neuronal function processes (Appendix A).

A greater resolution of these protein activities was gained through Ingenuity Pathway Analysis (IPA). IPA applies algorithms to establish a meaningful order and mechanistic links in a large biological data set based on published data and the output is presented as cellular functions (Table 1), intracellular pathways (Table 2), putative diseases (Table 3) and upstream regulators (Table 4). Cellular functions linked to the observed protein changes were related to synaptic transmission, e.g., synthesis of neurotransmitter, release/recycling/quantity of synaptic vesicles, size/quantity of postsynaptic density, loss/quantity of dendrites, mitochondria and to formation of amyloids (tau-filaments, Aß-plaques and other inclusion bodies, see Table 1). Pathways most prominently affected relate to neurotransmission and synapse integrity including axonal guidance signalling, p70S6K and PI3K/AKT signalling, mitochondrial function, NRF2-mediated stress responses, clathrin-mediated endocytosis and protein ubiquitination (see Table 2 and Appendix A). These proteins also play a role in AD, PD, cognitive impairment and neurodegeneration (Table 3). It is apparent from these tables that protein alterations observed for LMTM were largely not dose dependent for the dose range applied and further protein quantification analyses were conducted for the 15 mg/kg dose cohort.

### 3.1. Impact of Tau-Overexpression on the Neuroproteome in L66 Mice

The 123 protein species (71 spots from experiment 1) differentially regulated between the two genotypes were clustered for low expression in wild-type and heightened expression in L66 and vice versa (heatmap, Figure 4B). Sixty-three proteins clustered in the first category (red cluster in upper left quadrant), while 60 proteins clustered in the latter (green cluster in lower left quadrant). These clusters include 16 of the 27 shared protein species (intersection in the Venn diagram Figure 4A) which were increased in L66 (ATPA, HSP74, DHE3, GHC1, KPYM, ANXA7, PACN1, STK36, ACADV, ODPA, HPLN1, ADDB, HSP74, PCCA, NSF and IDHC, Figure 5A,B) and 11 protein species whose expression was lower in L66 (GTPC1, 2 species of RN181, AATM, MP2K1, DYN1, PYGB, PFKAL, ACON, RABE1 and HXK1, Figure 5A,B).

### 3.2. LMTM-Dependent Protein Expression in L66 Mice

In Experiment 2, two cohorts of L66 mice were exposed to LMTM (15 and 45 mg/kg daily for 8 weeks) and compared with a vehicle cohort. The 657 differentially regulated protein species (433 spots) were clustered according to their expression level in L66 (z-score created by heatmapper). The majority of these proteins (518 protein species = about 80%) showed high abundance in the vehicle cohort (see green cluster in the upper left quadrant of the heatmap Figure 4C), most of which were downregulated by both doses of LMTM. About half of these proteins were lowered to the average expression level (black); the other half was decreased below average expression (red). Other proteins (131 protein species = about 20%) were at a low level of expression for vehicle-treated mice (see red cluster in the lower left quadrant of the heatmap Figure 4C) but considerably upregulated in the 15 and 45 mg/kg LMTM cohorts (green).

Of particular interest are the 27 protein species (17 spots) that are dysregulated by tau-overexpression and modulated by LMTM treatment (intersection in the Venn diagram Figure 4A). Important are those proteins normalised after LMTM treatment (Figure 5A), which include mitochondrial glutamate dehydrogenase (DHE3), mitochondrial glutamate carrier (GHC1) and annexin 7 (ANXA7). Other proteins showed no clear direction in effect (e.g., isocitrate dehydrogenase, IDHC, and aspartate aminotransferase, AATM; Figure 5B).

Out of the 657 LMTM-regulated protein species only 26 were regulated in a dose-dependent fashion. These again clustered in three different regulation patterns: (i) stronger effect at higher dose (9 protein species were downregulated: DYN1, PP1R7, RN181, ENOA, HSP7C, HNRPC, HS74L, SAM50 and ATPA; and 2 upregulated: ATP5H and PPIA, Figure 5C); (ii) stronger effect at lower dose (5 protein species were downregulated: PABP1, ATPB, SUCA, HSP74 and ATPG; and 4 upregulated: HIBCH, ACTG, NECP1 and SPB6, Figure 5D); (iii) only 15mg/kg was effective (ATPB, ATPA, CACP, BPHL, SNAB and SNAA, Figure 5E). These contained not only heat shock proteins (e.g., HSP7C, HSP74), but also proteins with structural (DYN1, ACTG, SNAA) and metabolic/energetic functions (ATPA, ATPB, ATPG).

In addition, eight spots were shifted horizontally following LMTM treatment (Figure 5F), implying post-translational modifications (PTMs). NDUV1, KCRU, ALDOA, PSDE and RAM were shifted toward a basic isoelectric point (pH), while VDAC1, SODM and PRDX6 became more acidic, confirming the ability of LMTM to promote protein regulation at the functional level. Both SODM and PRDX6 are important scavengers for reactive oxygen species and integral parts of the NRF2 pathway (see below).

### 3.3. Tau-Dependent Pathway Rescue by LMTM

We have recently confirmed the enrichment of P301S tau in the synaptic compartment of L66 mice [41]. Further, our pathway analyses suggests that overexpression of tau leads to changes in synapses, and that these are modifiable by treatment with LMTM (Table 1, middle section). Accordingly, among the 27 protein species dysregulated by tau-overexpression and modulated by LMTM, several protein species have known functions in synaptic/neurotransmission, e.g., DYN1, ADDB, ANXA7 and PACN1 (Panther database). We therefore reverted to immunoblotting for quantification of selected proteins potentially responsive to treatment with LMTM (15 mg/kg) in L66 mice. Both α-synuclein and VAMP2 expression were reduced in L66 mice (Figure 6A,B; *p* < 0.05), but only α-synuclein levels were normalised in LMTM-treated cohorts (Figure 6C,D). PI3K/AKT involved in signal transduction and the mTOR kinase P70S6K were significantly altered in line with the IPA analyses (Table 2). This pathway is a known regulator of protein synthesis, cell survival and proliferation [67,68] and may induce the dysregulation of synaptic structural proteins. The activity of AKT as upstream regulator of mTOR was significantly lowered in L66 mice and both mTOR kinases P70S6K and 4E-BP1 followed a similar trend (Figure 7A,B). This may be explained in terms of a significant reduction in BDNF signalling and the lowered downstream kinase TrkB in L66 mice (Figure 7A,B and, additionally, Table 4 for BDNF), which impinge on the AKT pathway. Taken together, an overall significant decrease in the PI3K-AKT-mTOR pathway in L66 mice was confirmed (F(1,90) = 14.48; *p* = 0.0003). Intriguingly, all tau-related changes in the AKT-mTOR pathway were not normalised by LMTM according to immunoblotting (Figure 7C/D), a result which is at odds with our IPA analysis in which mTOR was activated following drug treatment (activation z-score = 2.1, see Table 4). This finding may be due to different sensitivity limits between the two methods. Since it is conceivable that the lack of trophic and structural support via BDNF and α-synuclein increases synapse/cell loss [69,70], we next sought to confirm this by quantitative counting of NeuN-immunoreactive neurons in tau-rich cortical areas and comparing genotypes and drug cohorts. Relative to wild-type, there was a reliable loss of principle cells in hippocampal CA1 (Figure 8A–C, F = 3.012, *p* = 0.040), CA3 (Figure 8D–F, F = 3.405, *p* = 0.027), inner (F = 3.869, *p* = 0.017) and outer granule cell layer (F = 11.44, *p* < 0.0001) of the dentate gyrus (Figure 8G–I), as well as in the motor cortex (Figure 8J–L, F = 43.75, *p* < 0.0001). This is likely related to the reduction in BDNF activity (Figure 7), as the cell loss was also not rescued despite relatively high doses of LMTM (45 mg/kg).

ERK and GSK-3 are crucial factors for neuronal survival and synaptic plasticity [71,72]. In L66 mice, phosphorylation of ERK at threonine202/tyrosine204 and GSK-3ß at serine9, were significantly increased and in CaMKII-α, which acts upstream of GSK-3, the phosphorylation at threonine286 followed a similar direction (Figure 9A,B); no phosphorylation changes of GSK-3α were detected. Exposure to LMTM only reversed pERK significantly (Figure 9C,D) but reliably increased the phosphorylation of GSK-3ß, further underlining the specificity of LMTM to normalise some but not all cell signalling cascades. It is reasonable to argue that failure to re-establish the CaMKII-α/GSK-3ß activities offers an explanation for continued cell loss in L66 mice even after LMTM treatment.

Given our recent observation that oligomeric tau is enriched in vesicular fractions derived from L66 mice [41], we explored mechanisms involved in release and recycling of synaptic vesicles. Proteins of interest included clathrin, dynamins (DYN1, DYN2 and DYN3), dynamin-like proteins (OPA1 and DNM1L), calcineurin (CANB1) and adaptor protein complex 2 (AP-2 or AP2M1), which were all differentially expressed between L66 and wild-type mice. Multiple proteins of this clathrin-mediated endocytosis signalling pathway (Appendix A) were regulated by LMTM (Table 2); dynamin-1, dynamin-3 and dynamin-like protein were all lowered, dynamin-1 and clathrin heavy chain were significantly increased. These results strongly suggest that LMTM is capable of normalising clathrin-mediated endocytosis and this mechanism might contribute to improved neurotransmission and behavioural performance in these mice [43,47].

### 3.4. Tau-Independent Pathway Rescue by LMTM

Stress-induced-phosphoprotein 1 (STIP1), glutathione S-transferase Mu 5 (GSTM5), glutathione S-transferase omega-1 (GSTO1), ubiquitin carboxyl-terminal hydrolase 14 (UBP14) and ferritin heavy chain (FRIH), were all increased in abundance by 1.3- to 2.2-fold in LMTM-treated cohorts. In addition, PTMs have been suggested for both the mitochondrial superoxide dismutase (SODM) and peroxiredoxin-6 (PRDX6) as they were shifted horizontally by LMTM, confirming its ability to promote protein regulation at a functional level (Figure 5F). These proteins are all part of the NRF2 pathway (Appendix A), which was both different between genotypes and responsive to LMTM treatment (Table 2). Moreover, there is considerable crosstalk between NRF2 and mitochondria to maintain mitochondrial redox homeostasis by providing important reactive oxygen species scavengers such as glutathione peroxidase, glutathione reductase, SODM and peroxiredoxins [73]. It is therefore not surprising that the expression of proteins related to mitochondrial function was altered in L66 and recovered following LMTM treatment (Table 2). This included mitochondrial complex I enzyme ETFD and complex III enzymes ATPA and ATPG the abundance of which were increased in L66 compared to wild-type. LMTM seems to have a great impact on mitochondria (Appendix A). LMTM acted mainly on complexes I, III and V of the electron transport chain (ETC) in L66 mice but it also reduced numerous enzymes involved in the TCA cycle and in fatty acid metabolism, e.g., AATM, ACADV, ACLY, BACH, ECHA, ECHB, THIM, ACON, DHE3, MDHM, ODO2 and SUCA in support of mitochondrial recovery of function. In summary, LMTM induced the NRF2-mediated antioxidant defence system and appeared to facilitate energy biogenesis and mitochondrial function as suggested for the parent compound MTC [51,57].

Although not regulated between genotypes (Table 2), protein ubiquitination was strongly modified by LMTM in L66 mice (Appendix A). Ubiquitination includes several heat shock proteins (HSP), proteasome regulatory subunits (PSM), ubiquitin-protein ligases and multiple chaperones that were all modulated by LMTM. Although their role in AD and FTD is uncertain, chaperones like HSP74 and HS105 were decreased, while STIP1 and HSP7C were increased suggesting that protein degradation is facilitated by LMTM treatment. For the E3 ubiquitin-protein ligase RNF181, a RING-type E3 ubiquitin transferase with a total of at least 23 proteins, species 8, 9 and 200 were increased more than 1.5-fold (Appendix A). Three further RNF181 species (260, 283 and 305) were lowered by LMTM, and they contained tau at very low abundance (between 0.5 and 4% of total spot volume, data not shown). These results support the notion that LMTM induces or facilitates the degradation and removal of aggregated tau.

## 4. Discussion

In this study, we performed an unbiased proteomic investigation comparing brain samples of wild-type and L66 tau-transgenic mice and assessed the ability of LMTM in reversing these changes in L66. L66 harbours the FTD-associated mutation P301S in the *MAPT* gene. These mice show early onset high tau load in hippocampal and cortical neurons, a normal cholinergic phenotype, a robust neuroinflammation and sensorimotor and motor learning deficits [40,42]. Several of these phenotypes, including tau pathology and behavioural deficiencies, were reversed by treatment with LMTM [43,47]. We have now extended these studies using a similar LMTM regime in L66 mice to identify key pathways underlying these corrections. Since we have used whole brain extracts without region-specificity, the alterations that we report for the brain proteome may have occurred in neurons, astrocytes or both [38]. However, the changes are most likely associated with neurons, since these cells account for around two-third of the different cell types in the mouse brain [74], tau expressed in L66 mice is under the neurone-specific regulatory element Thy1 [40], and tau aggregates are most prevalent in neurons in mutant tau mice [75] and human tauopathies [76]. A limitation of this study, however, is that the resulting proteomic analyses do not provide spatial resolution of FTD-related brain areas but rather reflect changes that occur across the whole brain. in contrast to the recently preferred LC-MS/MS techniques, we have separated protein species before digestion, identification and quantification by 2-DE gel technology (for comparison see [24,77]), as it offers great flexibility and tunable resolution, and enables the simultaneous analysis of the total protein, protein species and post-translational modifications in large-scale studies including changes with functional relevance to disease mechanisms [66,78,79,80].

The key findings that we report here are that mutant tau induced metabolic/mitochondrial dysfunction, changes in synaptic transmission and stress responses and that LMTM recovered these functions in a dose-independent manner. LMTM-activated pathways not affected by tau-overexpression included NRF2, oxidative phosphorylation and protein ubiquitination. In contrast, LMTM did not recover trophic support and neuronal cell loss induced by the expression of mutant tau.

### 4.1. L66 Mice Show Common Pathways Dysregulated in FTD Patients

Only a few proteomic studies are available for FTD (for review see [21,22]) and many show a large overlap between FTD-tau, FTD-TDP43, AD and other neurodegenerative disorders. Hu et al. identified 151 differentially regulated proteins, but these only yielded 78% specificity for FTD-TDP43 over FTD-tau [26]. Others reported 56 differentially regulated proteins and only 10 of these were different between the FTD-TDP43 and FTD-tau sub-types [27]. Even fewer putative biomarkers (15) were revealed in a comparative study of AD and FTD patients with only seven of these differentiating between both dementias [28] suggesting that it will be difficult to achieve specificity in protein biomarkers for any one form of dementia. This appears to hold, since spinal cord from FTD patients contained 52 dysregulated proteins compared to controls, but 33 of these were shared between FTD and amyotrophic lateral sclerosis (ALS) and played a role in mitochondrial dysfunction and metabolic impairment [23]. By contrast, disrupted protein synthesis and vesicle trafficking [81], and synaptic transmission and inflammation [82] specifically changed in FTD brains compared with progressive supranuclear palsy or ALS emphasising that different alterations in protein content and activity might still be determined when studying very selective dementia cohorts. Proteomics studies in experimental FTD models are limited too. Using 2D-DIGE to investigate spinal cord tissue from 7-month-old P301S tau mice, the antioxidant proteins peroxiredoxin-6, heat shock protein 2, apolipoprotein E and latexin were all enhanced in astrocytes underlining a neuroprotective function of these cells [38,39]. Critically, few dysregulated protein network clusters including metabolic/mitochondrial, synaptic transmission and stress responses seem to be commonly disrupted in experimental and clinical FTD studies and indeed, these networks were amongst those dysregulated in L66 and modified by LMTM.

### 4.2. Tau Accumulation in L66 Mice Affects Multiple Brain Proteome Networks

In our L66 mice, tau overexpression revealed a significant differential regulation of about 3.5% of all spots (Experiment 1: 71 regulated spots out of around 2000 visible spots). Of the dysregulated protein species, approximately 30% were metabolic, 13% belonged to neurotransmission/neurodevelopment, while 9% were structural proteins and 12% played a role in oxidative stress. Dysregulation of metabolic processes in general [83] and mitochondrial function in particular [84] have been reported before for the brain proteome of P301L tau mice. Immunoprecipitation combined with mass spectroscopy also revealed a reduction in mitochondrial function, specifically in proteins involved in the ETC, together with an increase in heat shock proteins. These findings correlated with the amount of tau load in neurons ([39]; this study) and are clearly in line with clinical investigations underpinning the suitability of P301L/P301S mutant mice as a model for FTD. A presynaptic location for these changes is further indicated by a decrease in amphiphysin-1 levels [85], a protein involved in clathrin-mediated endocytosis [86,87,88], and in L66 mice were most likely due to the accumulation of non-phosphorylated tau in the pre-synapse [41]. A complementary study on RNA profiling conducted in young (2 months) and old (12 months) P301S mice found that most of the differentially regulated genes were associated with synaptic signalling [75]. In their study, gene-networks linked to pre-synaptic vesicle trafficking were altered throughout disease progression, while gene-networks linked to the post-synapse, i.e., glutamate signalling, were dysregulated in young but not in old mice. We only investigated 6-month-old L66 mice but confirmed that levels of proteins for both synaptic vesicle trafficking (e.g., endocytosis, transport and recycling of synaptic vesicles, see Table 1) and glutamate signalling (degradation and biosynthesis of glutamate, see Table 2) were altered. Further, a recent study established a negative correlation between tau and glutamate receptor activity in FTD patients with tau mutations [89]. Taken together, these results provide compelling evidence that metabolic dysfunction, synaptic transmission and stress response pathways are a consistent phenotype associated with overexpression of tau mutated at residue 301, and these changes match those frequently reported in clinical studies. L66 mice, therefore, are an ideal translational model to examine drug effects in preclinical conditions.

### 4.3. LMTM Corrects Metabolic Dysfunction, Synaptic Transmission and Stress Responses in L66 Mice

LMTM is being developed as a treatment targeting pathological aggregation of tau protein in AD and FTD [48,49,90,91], and this strategy appears promising since a correlation between the degree of tau aggregation and the severity of dementia has been established [12,13,37,45]. LMTM blocks tau aggregation in vitro [46,92], reverses behavioural deficits in tau transgenic mice [47] and preserves cognitive decline and brain atrophy in AD patients [48,49]. In addition, we have recently reported that LMTM restores acetyltransferase immunoreactivity in basal forebrain neurons, increases hippocampal acetylcholine levels and complex IV activity in a mouse model of AD and normalises glutamate release from synaptosomal preparations ex vivo [43,56]. We have previously reported significant reductions in tau pathology and the correction of behavioural deficiencies in L66 mice using a similar LMTM dosing regimen [47]. The main findings from the latter study were the slightly higher overall reduction of tau-positive neurons for the 45 mg/kg dose of LMTM, while behavioural correction of the motor phenotype on the rotating road was fully recovered following administration of the lower 15 mg/kg dose (see Figures 7 and 9 in [47]). Our proteomics investigation, that also included 15 and 45 mg/kg LMTM, revealed that 22% of all spots (433/2000) were regulated by LMTM and the majority of these (95%) followed the same regulation pattern for both doses. These findings are in line with behavioural analyses of L66 and with a recent pharmacokinetic study in AD and FTD patients showing that the maximal effective dose for LMTM is at 16 mg per day for AD and 20–60 mg per day for FTD, while higher doses are devoid of additional benefit [47,48,49].

In terms of dysregulated protein species, roughly 30% were metabolic, 14% belong to neurotransmission/neurodevelopment, 23% were involved in protein transcription (transcription/translation/biosynthesis/modification/degradation), while 6% played a role in oxidative stress. From our pathway analyses, metabolic dysfunction, synaptic transmission, and stress responses were altered due to tau expression and corrected/improved by LMTM, underlining the pluripotent effect of the drug. While we have no confirmatory evidence, we propose that effects of LMTM are most likely mediated indirectly by its mechanism as a tau aggregation inhibitor as these pathways were altered by overexpression of mutant tau. By contrast, other pathways that could mitigate detrimental effects of tau, but are presumably mediated directly and not via tau dissolution are, for example, NRF2, oxidative phosphorylation and protein ubiquitination (see below).

### 4.4. No Effect of LMTM on Trophic Support Networks

BDNF, which is known to promote neuronal survival, was reduced in L66 mice, in line with reports in several neurodegenerative disorders in humans, including FTD [93]. In P301S mice, a BDNF-related mechanism by which tau induces neuronal dysfunction has been established [94], while others have reported increased BDNF levels specifically in astrocytes [75]. In L66, neither the reduced trophic support through BDNF nor the neuronal degeneration were affected by LMTM. This clearly establishes specificity for the mechanistic correction of LMTM although the reported MTC increase in BDNF in dopaminergic neurons [95] was not specifically addressed here. A dose comparison between our study and the work of Bhurtel and colleagues, however, confirmed that equivalent doses had been used, but regimes and length of administration were different. Furthermore, reduced trophic support through BDNF is often connected to a reduction in mTOR activity; dysregulation (both, increase and decrease) of mTOR and associated pathways has been widely reported in neurodegenerative disorders [96,97]. Our pathway analyses highlighted significant alterations of BDNF/mTOR and downstream PI3K-AKT/70S6K (Table 4 and Table 2, respectively) and so the latter pathway was investigated in more detail. In L66, PI3K-AKT-mTOR and downstream kinases p70S6K and 4E-BP1 were decreased and ERK-GSK-3ß was increased. And although a small reversal of ERK overactivation was achieved, no effect on 70S6K and 4E-BP1 were noted following LMTM treatment. It appears, therefore, that this rescue of ERK, or any cellular pathways other than BDNF that can lead to mTOR activation [98,99], is unlikely to be a principle target for LMTM.

### 4.5. Tau Compromises Presynaptic Structures and Their Recovery by LMTM

Of particular interest are actions of LMTM on synaptic transmission and mitochondrial function (metabolism and stress). There were considerable changes in these categories in L66 mice related to the overexpression of mutant tau. These pathways are tightly connected [100,101] and mitochondria have an important role in the homeostatic regulation of presynaptic function [102]. It has been shown recently that tau protein is bound to the mitochondrial outer membrane and also enters the intermembrane space of mitochondria [103]. PHF-tau can form complexes with the voltage-dependent anion selective channel protein (VDAC) in the mitochondrial outer membrane, and also with ATP synthase subunit 9 and core protein 2 of complex III in the intermembrane space [104]. These interactions are likely to be deleterious for electron transport in mitochondria, and we suspect that LMTM recovers mitochondrial function by direct interference with oxidative phosphorylation; most likely, LMTM would serve as a redox cycler similar to its parent compound MTC [52,53,105]. Furthermore, tau binding promotes presynaptic actin polymerisation to crosslink synaptic vesicles, which strongly restricts their mobilisation and affects docking, transport and fusion of synaptic vesicles and impairs neurotransmitter release [88]. LMTM is predicted to normalise pre-synaptic structure/function by dissolution of oligomeric tau enriched in the pre-synaptic compartment of L66 mice [41], and this may be aided by the induction of clathrin-mediated endocytosis and protein ubiquitination. Although LMTM did not seem to correct expression levels of individual synaptic structural proteins, an overall functional rescue of synaptic function by LMTM has been established [43,56].

### 4.6. LMTM Promotes Antioxidant Response via NRF-2 Activation in a Tau-Independent Fashion

The ability of MTC to activate cellular defence mechanisms by activation of NRF2 has been revealed in vitro [106] and in ageing, and it is associated with the restoration of deficient antioxidant defence proteins such as NRF1 and SOD2 [51]. In P301S mice, MTC also reduced glutathione-induced stress, increased SODM (or SOD2) activity and activated NRF2-mediated antioxidant responses [57]. We here confirm, in L66 mice, that proteins downstream of NRF2 such as stress-induced-phosphoprotein 1 (STIP1), glutathione S-transferase Mu 5 (GSTM5), glutathione S-transferase omega-1 (GSTO1), ubiquitin carboxyl-terminal hydrolase 14 (UBP14), ferritin heavy chain (FRIH) and mitochondrial superoxide dismutase (SODM), were increased up to more than 2-fold by LMTM. LMTM therefore, similar to MTC, promotes cellular defence by activating the NRF2 system.

## 5. Conclusions

The mutant form of tau in L66 mice induced protein changes related to metabolic/mitochondrial function, synaptic transmission and stress responses. These functions are common to transgenic models and clinical FTD studies. LMTM recovered these functions in a dose-independent manner and confirmed its effect on tau-related dysfunction. Furthermore, LMTM has activated pathways that were unaffected by tau-overexpression. These included NRF2 antioxidant response, oxidative phosphorylation and protein ubiquitination, but did not show activity towards functions involved in trophic support and neuronal degeneration.

## Figures and Tables

**Figure 1 cells-10-02162-f001:**
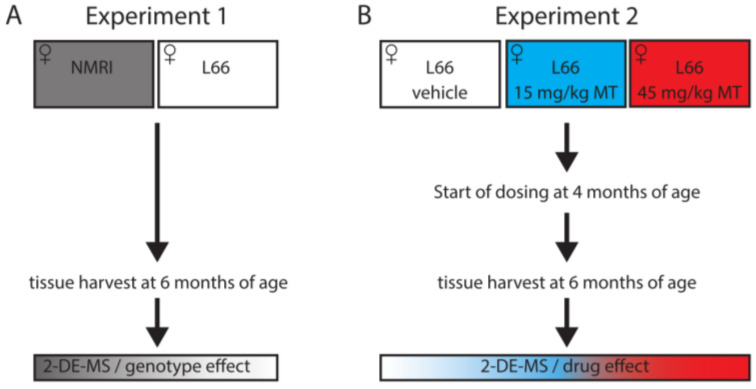
Study design for proteomics analyses. Proteomics analyses were conducted in (**A**) untreated wild-type and L66 mice, to identify proteins changing due to tau over-expression (experiment 1), and (**B**) in vehicle- or LMTM-treated L66 mice, to identify the effect of LMTM (experiment 2).

**Figure 2 cells-10-02162-f002:**
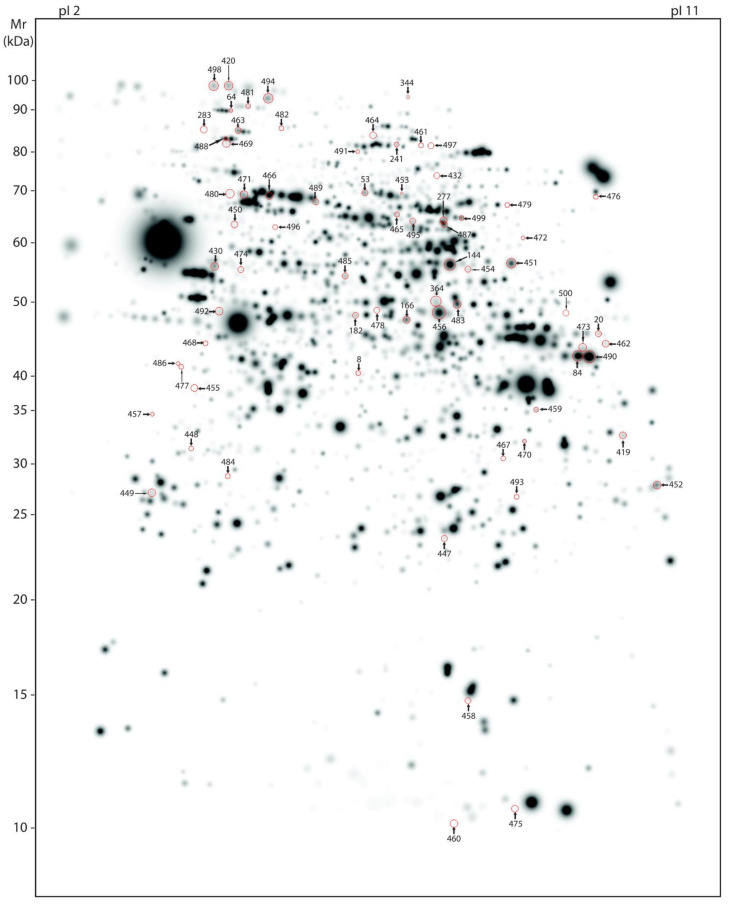
Master gel representing protein spots differentially abundant between genotypes. Average, large-scale 2-DE gel pattern of brain proteins extracted with urea and stained with silver. All 71 protein spots varying between the genotypes (wild-type and L66) were identified by MS and are highlighted by arrows in representative wild-type gel (artificial gel image created by the Proteomweaver software that shows average spot intensities of the wild-type mouse cohort). Identification numbers (spot IDs) correspond to numbering as provided in the Appendix A.

**Figure 3 cells-10-02162-f003:**
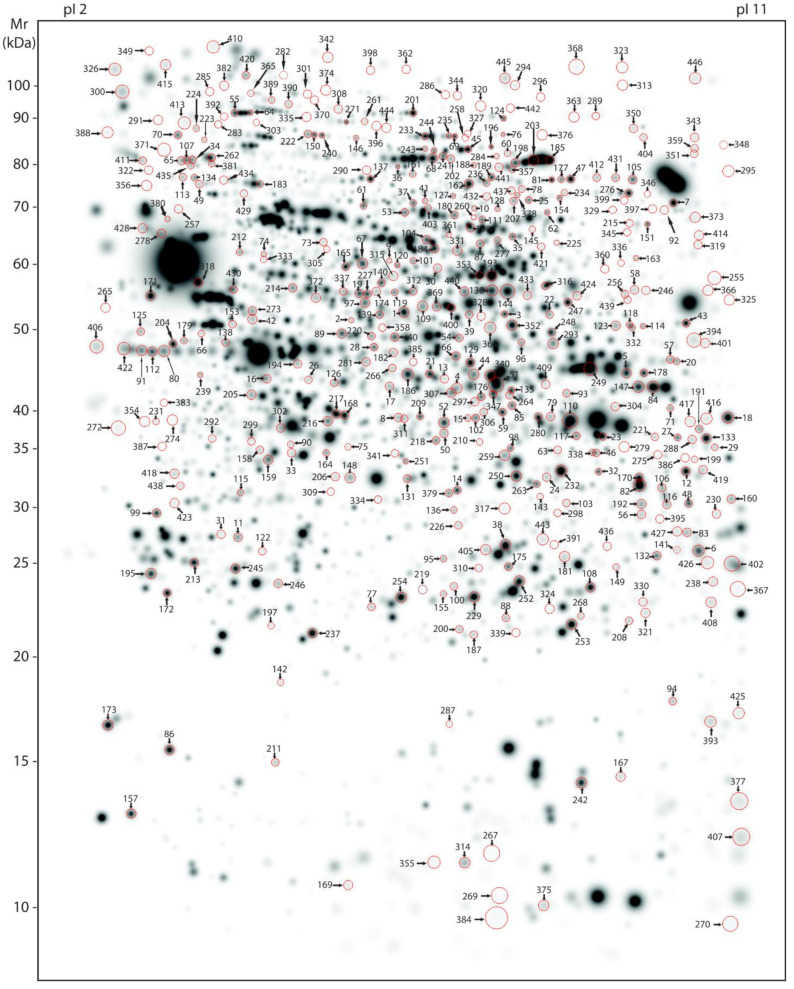
Master gel representing protein spots regulated by LMTM. Average, large-scale 2-DE gel pattern of brain proteins extracted with urea and stained with silver. All 433 protein spots varying due to LMTM treatment (L66 treated with either vehicle or different doses of LMTM) were identified by MS and are highlighted by arrows in a representative vehicle-treated L66 gel (artificial gel image created by the Proteomweaver software that shows average spot intensities of the vehicle-treated L66 mouse cohort). Identification numbers (spots IDs) correspond to numbering as provided in the Appendix A.

**Figure 4 cells-10-02162-f004:**
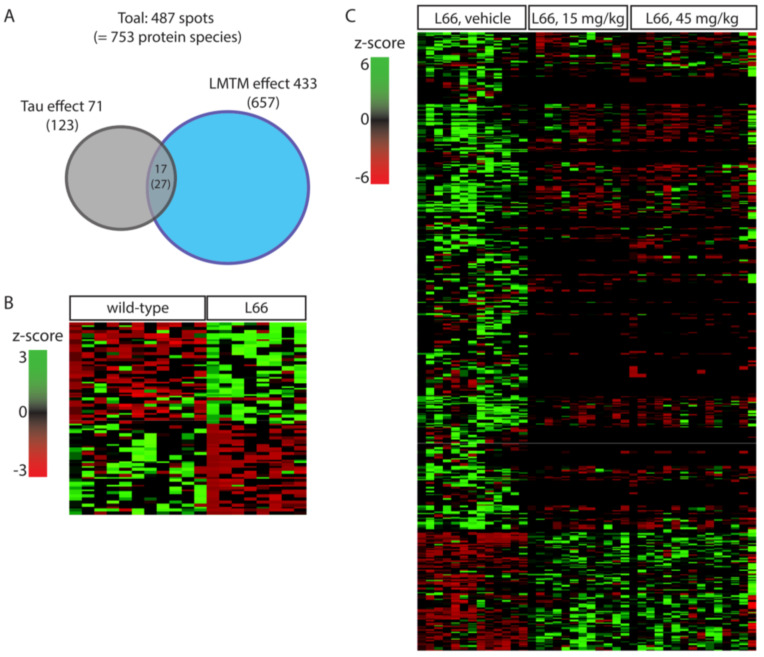
Spots and protein species differentially abundant between wild-type and L66 mice and regulated by LMTM. (**A**) Venn diagram with overview of regulated proteins showing 71 spots (123 protein species) with different expression between genotypes and 433 spots (657 protein species) regulated by LMTM (15 and 45 mg/kg). Seventeen spots (27 protein species) were common to both analyses. (**B**) Heatmap for protein species density in L66 compared to wild-type control mice. (**C**) Heatmap for protein species density in LMTM-treated L66 mice compared to vehicle-treated L66 mice.

**Figure 5 cells-10-02162-f005:**
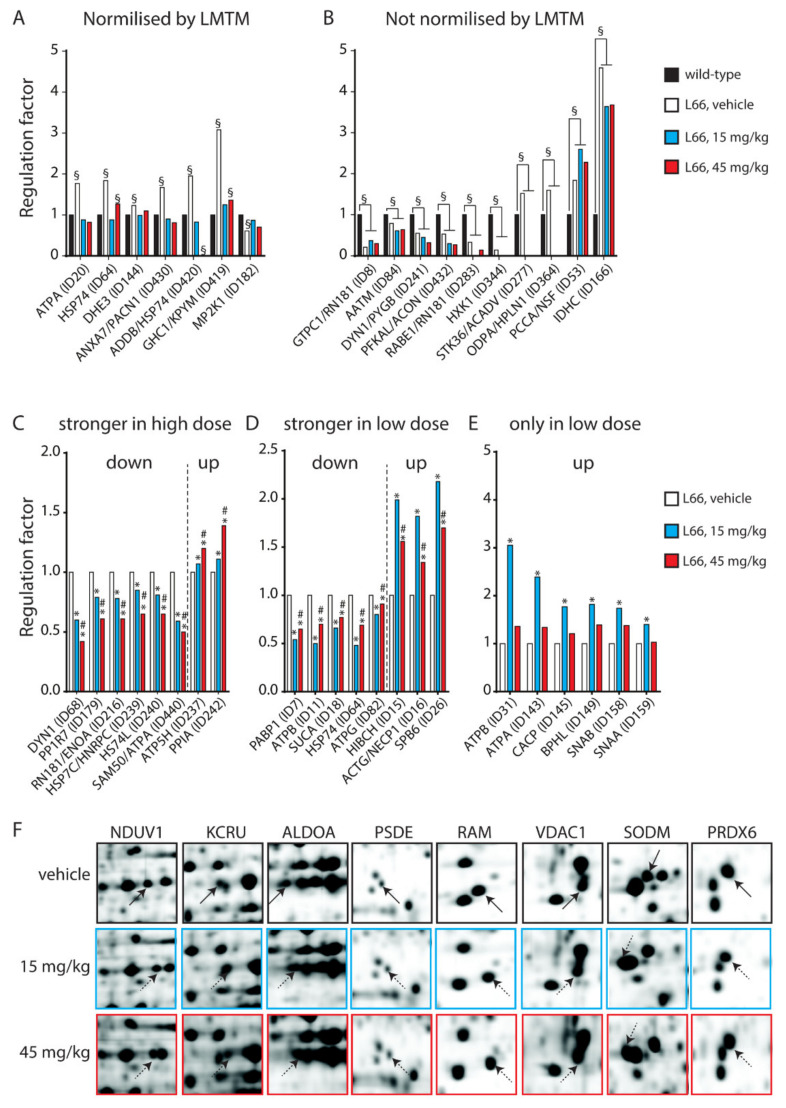
Protein species with LMTM dose-dependent regulation. (**A**,**B**) Twenty-seven protein species (17 spots) were common between experiments 1 and 2. (**A**) Ten of these protein species (7 spots) were normalised by LMTM, (**B**) while no clear effect was identified for the remaining (17 species in 10 spots). (**C**–**E**) Twenty-six protein species (22 spots) were dose-dependently regulated with (**C**) stronger effect at higher dose (9 protein species downregulated and 2 upregulated), (**D**) stronger effect at lower dose (5 protein species downregulated and 4 upregulated), (**E**) only 15mg/kg was effective (6 protein species upregulated). (**F**) Eight further protein species were regulated at the functional level as they shifted horizontally following LMTM treatment, implying post-translational modifications. §: *p* < 0.05 vs. wild-type, *: *p* < 0.05 vs. L66-vehicle, #: *p* < 0.05 vs. L66-15LMTM.

**Figure 6 cells-10-02162-f006:**
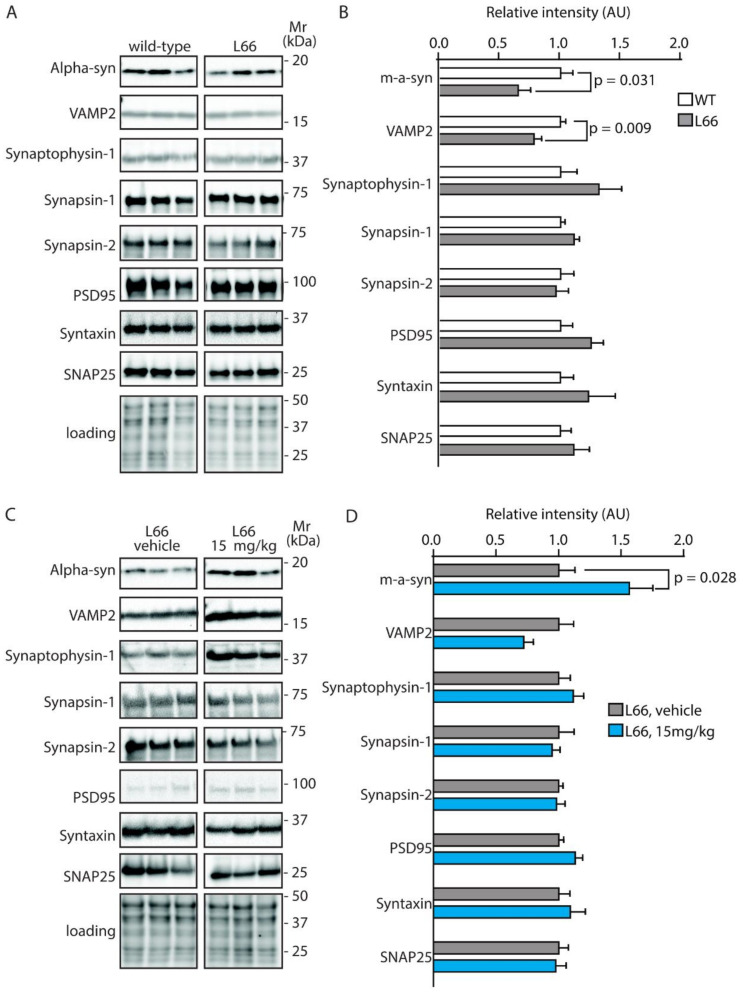
Quantification of synaptic proteins in mice. Representative immunoblots for (**A**) wild-type and L66 mice and (**C**) L66 mice treated with vehicle or LMTM (15 mg/kg). The proteins were separated by Tris-glycine SDS-PAGE and immuno-detection conducted using the antibodies specified. (**B**,**D**) Densitometric quantification of these proteins was normalised to total protein loading and values expressed as group mean ± S.E. Statistical analyses were conducted using unpaired t-test for n = 8–12 (**B**) and n = 11–14 mice (**D**) per group. WT: wild-type. Original blot images are provided in the Appendix A.

**Figure 7 cells-10-02162-f007:**
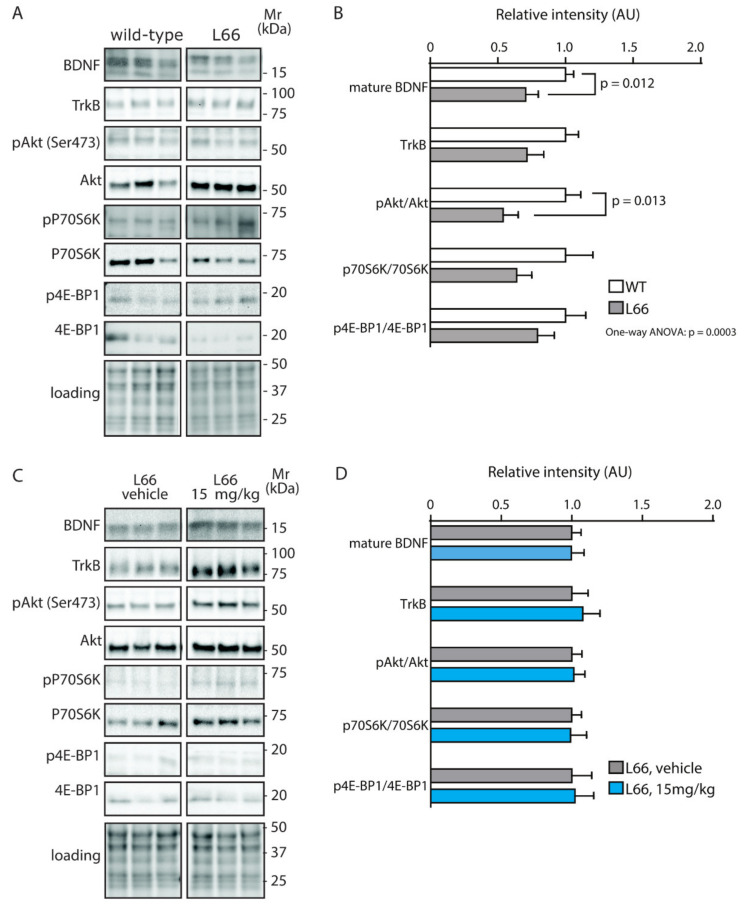
Quantification of BDNF/PI3K-AKT-mTOR activity in mice. Representative immunoblots for (**A**) wild-type and L66 mice and (**C**) L66 mice treated with vehicle or LMTM (15 mg/kg). The proteins were separated by Tris-glycine SDS-PAGE and immuno-detection conducted using the antibodies specified. (**B**,**D**) Densitometric quantification of these proteins was normalised to total protein loading and values expressed as group mean ± S.E. Statistical analyses were conducted using unpaired t-test and one-way ANOVA for n = 8–12 (**B**) and n = 11–14 mice (**D**) per group. WT: wild-type. Original blot images are shown in the Appendix A.

**Figure 8 cells-10-02162-f008:**
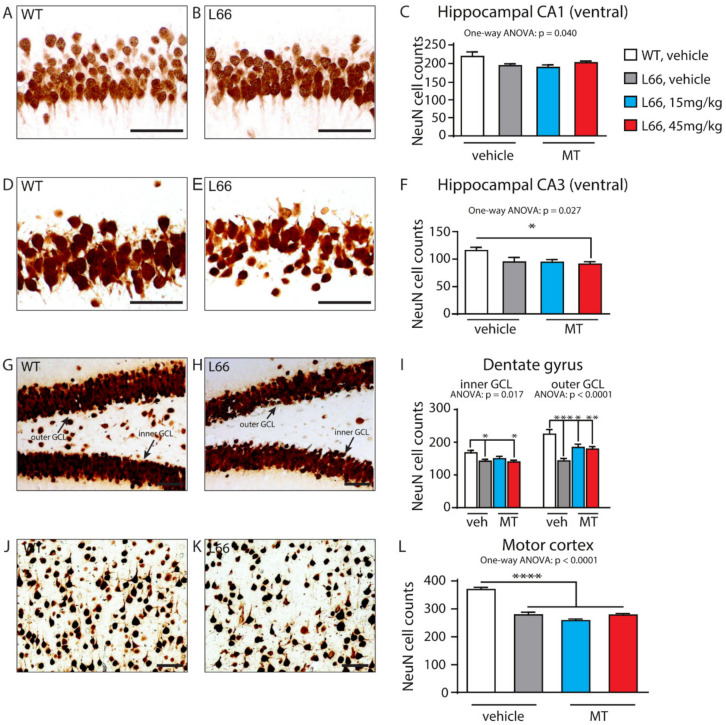
Neuronal cell counts between genotypes and in drug cohorts. Quantification of NeuN-immunoreactive neurons by manual cell counting in hippocampal CA1 (**A**–**C**), hippocampal CA3 (**D**–**F**), the inner and outer granule cell layers of the dentate gyrus (**G**–**I**), as well as in motor cortex (**J**–**L**) in wild-type (**A**,**D**,**G**,**J**) and L66 (**B**,**E**,**H**,**K**) mice. Loss of neurons in L66 mice was evident in hippocampal CA1 (**C**, *p* = 0.040) and CA3 areas (**F**, *p* = 0.027), in inner and outer GCL of the dentate gyrus (**I**, *p* = 0.017 and *p* < 0.0001, respectively), as well as in the motor cortex (**L**, *p* < 0.0001). LMTM did not reverse this cell loss in either area (**C**,**F**,**I**,**L**). Values are expressed as group mean ± S.E. Statistical analyses were conducted using one-way ANOVA and Bonferroni corrected t-test for n = 9–13 mice per group. *: *p* < 0.05, **: *p* < 0.01, ***: *p* < 0.001, ****: *p* < 0.0001 compared to vehicle-treated wild-type mice. WT: wild-type.

**Figure 9 cells-10-02162-f009:**
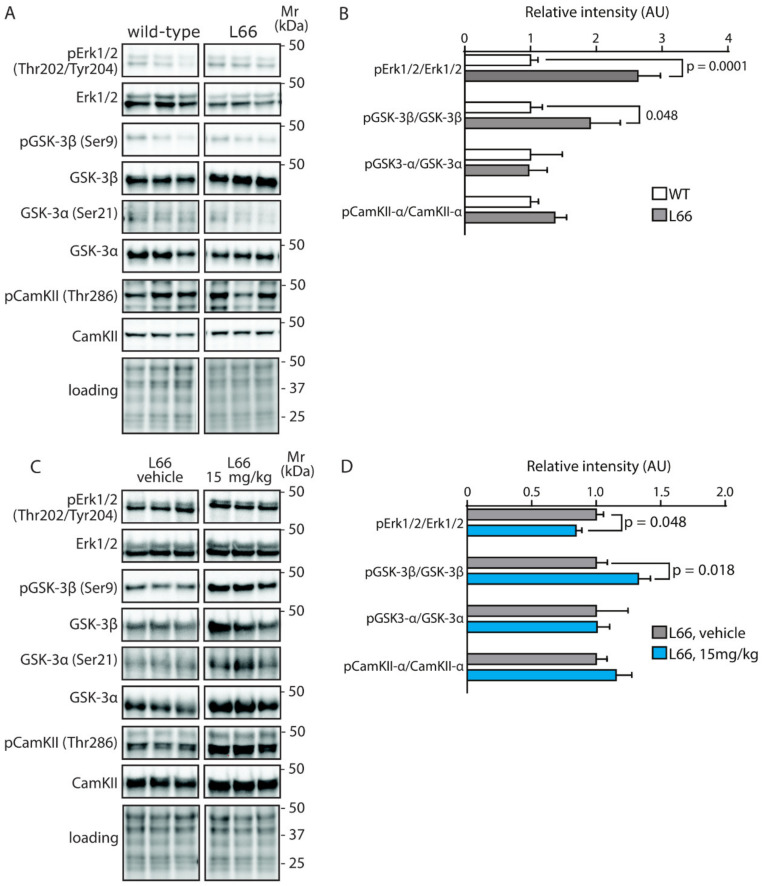
Quantification of ERK, GSK3 and CaMKII activity in mice. Representative immunoblots for (**A**) wild-type and L66 mice and (**C**) L66 mice treated with vehicle or LMTM (15 mg/kg). The proteins were separated by Tris-glycine SDS-PAGE and immuno-detection conducted using the antibodies specified. (**B**,**D**) Densitometric quantification of the various proteins was normalised to total protein loading and values expressed as group mean ± S.E. Statistical analyses were conducted using unpaired t-test for n = 8–12 (**B**) and n = 11–14 mice (**D**) per group. WT: wild-type. Original blot images are shown in the Appendix A.

**Table 1 cells-10-02162-t001:** Summary of significant hits for cellular functions identified using IPA Core Analysis. The cellular functions are significantly altered due to tau overexpression, to LMTM treatment or both. Significant alterations were found by comparing our proteomics data with the Ingenuity Knowledge Base. The cellular functions are named and corresponding *p*-values are given.

		*p*-Value		
	FUNCTIONS	L66 vs.	L66 (15mg/kg MT)	L66 (45mg/kg MT)
		Wild-Type Controls	vs. L66 (Vehicle Control)	vs. L66 (Vehicle Control)
Genotype	Patterning of dendrites	3.63 × 10^−4^	-	-
	Abnormal quantity of synaptic vesicles	7.08 × 10^−4^	-	-
	Long-term potentiation	6.31 × 10^−3^	-	-
	Quantity of dendrites	7.08 × 10^−3^	-	-
	Excitation of spiny neurons	1.20 × 10^−2^	-	-
	Quantity of postsynaptic density	1.20 × 10^−2^	-	-
	Elongation of axons	1.70 × 10^−2^	-	-
	Synthesis of dopamine	1.82 × 10^−2^	-	-
	Synthesis of l-glutamic acid	1.82 × 10^−2^	-	-
	Loss of hippocampal neurons	3.63 × 10^−2^	-	-
	Maturation of synaptic vesicles	3.63 × 10^−2^	-	-
	Release of neurotransmitter	4.57 × 10^−2^	-	-
Genotype and LMTM	Endocytosis of synaptic vesicles	6.31E x 10^−7^	7.94 × 10^−7^	8.71 × 10^−8^
	Transport of synaptic vesicles	2.82 × 10^−6^	3.47 × 10^−6^	8.91 × 10^−7^
	Morphology of presynaptic terminals	7.41 × 10^−5^	1.86 × 10^−2^	2.09 × 10^−2^
	Recycling of synaptic vesicles	3.47 × 10^−4^	6.92 × 10^−5^	7.24 × 10^−6^
	Density of dendritic spines	1.20 × 10^−2^	2.95 × 10^−3^	4.79 × 10^−3^
	Synaptic transmission of neurons	1.45 × 10^−2^	6.76 × 10^−4^	1.20 × 10^−3^
	Size of postsynaptic density	1.82 × 10^−2^	3.47 × 10^−4^	4.27 × 10^−4^
	Neurotransmission	2.82 × 10^−2^	7.41 × 10^−6^	2.29 × 10^−5^
	Loss of dendrites	3.02 × 10^−2^	3.39 × 10^−3^	4.07 × 10^−3^
	Synaptic depression	4.17 × 10^−2^	5.62 × 10^−4^	1.12 × 10^−3^
	Synaptic transmission	4.27 × 10^−2^	3.39 × 10^−6^	1.00 × 10^−5^
LMTM	Size of neurons	-	1.74 × 10^−5^	3.80 × 10^−5^
	Association of synaptic vesicles	-	3.47 × 10^−4^	4.27 × 10^−4^
	Formation of cellular inclusion bodies	-	6.61 × 10^−4^	1.15 × 10^−4^
	Neurodegeneration of axons	-	9.77 × 10^−4^	1.74 × 10^−3^
	Swelling of mitochondria	-	2.04 × 10^−3^	2.45 × 10^−3^
	Microtubule dynamics	-	2.51 × 10^−3^	5.01 × 10^−3^
	Size of axons	-	3.39 × 10^−3^	4.07 × 10^−3^
	Depolarisation of mitochondria	-	3.39 × 10^−3^	4.07 × 10^−3^
	Abnormal morphology of axons	-	6.03 × 10^−3^	3.55 × 10^−2^
	Production of reactive oxygen species	-	8.71 × 10^−3^	1.15 × 10^−2^
	Formation of amyloid-beta plaques	-	1.15 × 10^−2^	1.38 × 10^−2^
	Formation of tau filament	-	1.86 × 10^−2^	2.09 × 10^−2^
	Ubiquitination of protein	-	1.86 × 10^−2^	2.09 × 10^−2^

**Table 2 cells-10-02162-t002:** Summary of significant hits for canonical pathways identified using IPA Core Analysis. The pathways are significantly altered due to tau overexpression, to LMTM treatment or both. Significant alterations were found by linking our proteomics data with the Ingenuity Knowledge Base. The pathways linked to protein changes are named and corresponding *p*-values are given. The highlighted pathways (bold) are of specific interest and are discussed in detail in the text and in Appendix A.

		*p*-Value		
	PATHWAYS	L66 vs.	L66 (15mg/kg MT)	L66 (45mg/kg MT)
		Wild-Type Controls	vs. L66 (Vehicle Control)	vs. L66 (Vehicle Control)
Genotype	Protein kinase A signalling	3.89 × 10^−4^	-	-
	GnRH signalling	5.50 × 10^−3^	-	-
	Gap junction signalling	1.07 × 10^−2^	-	-
	Aspartate biosynthesis	1.82 × 10^−2^	-	-
	IGF-1 signalling	2.04 × 10^−2^	-	-
	Cell cycle	2.34 × 10^−2^	-	-
	Synaptic long term potentiation	2.45 × 10^−2^	-	-
	Wnt/Ca^2+^ pathway	2.45 × 10^−2^	-	-
	fMLP signalling in neutrophils	2.95 × 10^−2^	-	-
	ERK5 signalling	4.68 × 10^−2^	-	-
Genotype and LMTM	Gluconeogenesis I	3.63 × 10^−6^	4.90 × 10^−10^	2.45 × 10^−11^
	Glycolysis I	1.66 × 10^−4^	1.62 × 10^−13^	4.47 × 10^−7^
	TCA cycle II (eukaryotic)	1.66 × 10^−4^	1.00 × 10^−11^	2.45 × 10^−11^
	P70S6K signalling	5.13 × 10^−4^	9.55 × 10^−3^	1.51 × 10^−2^
	**Mitochondrial function**	1.48 × 10^−3^	9.55 × 10^−12^	5.50 × 10^−11^
	Huntington’s disease signalling	1.86 × 10^−3^	1.12 × 10^−6^	8.91 × 10^−7^
	Unfolded protein response	2.69 × 10^−3^	1.05 × 10^−2^	1.48 × 10^−2^
	**Clathrin-mediated endocytosis signalling**	2.82 × 10^−3^	6.76 × 10^−7^	6.46 × 10^−8^
	14-3-3-mediated signalling	4.07 × 10^−3^	1.86 × 10^−3^	3.31 × 10^−3^
	Glutamate biosynthesis II	1.20 × 10^−2^	3.72 × 10^−2^	4.07 × 10^−2^
	**NRF2-mediated oxidative stress response**	1.66 × 10^−2^	3.98 × 10^−6^	1.00 × 10^−8^
	Glutamate degradation II	1.82 × 10^−2^	1.02 × 10^−3^	1.26 × 10^−3^
	PI3K/AKT signalling	2.51 × 10^−2^	1.26 × 10^−2^	5.50 × 10^−3^
	Regulation of EIF4 and P70S6K signalling	4.07 × 10^−2^	2.00 × 10^−2^	3.09 × 10^−2^
	Tight junction signalling	5.00 × 10^−2^	4.27 × 10^−2^	2.29 × 10^−2^
LMTM	Aspartate degradation II	-	5.89 × 10^−7^	8.71 × 10^−7^
	Oxidative phosphorylation	-	3.55 × 10^−6^	8.51 × 10^−6^
	**Protein ubiquitination pathway**	-	9.33 × 10^−5^	6.61 × 10^−5^
	RhoA signalling	-	9.55 × 10^−5^	3.63 × 10^−5^
	Parkinson’s signalling	-	1.82 × 10^−4^	2.63 × 10^−4^
	Fatty acid α-oxidation	-	6.61 × 10^−4^	1.00 × 10^−2^
	NADH repair	-	1.02 × 10^−3^	1.26 × 10^−3^
	Signalling by Rho family GTPases	-	1.20 × 10^−3^	2.24 × 10^−4^
	PPARA/RXRA activation	-	1.78 × 10^−3^	1.12 × 10^−2^
	Glutathione-mediated detoxification	-	3.02 × 10^−3^	4.92 × 10^−7^
	Telomerase signalling	-	4.57 × 10^−3^	7.41 × 10^−3^
	Super pathway of methionine degradation	-	6.76 × 10^−3^	8.71 × 10^−3^
	Regulation of actin-based motility by Rho	-	1.12 × 10^−2^	3.63 × 10^−3^
	Semaphorin signalling in neurons	-	1.32 × 10^−2^	1.82 × 10^−2^
	VEGF signalling	-	1.32 × 10^−2^	4.47 × 10^−3^
	Neuregulin signalling	-	1.38 × 10^−2^	2.04 × 10^−2^
	Axonal guidance signalling	-	1.70 × 10^−2^	1.66 × 10^−2^
	PPAR signalling	-	1.70 × 10^−2^	2.45 × 10^−2^
	Super pathway of citrulline metabolism	-	1.70 × 10^−2^	2.09 × 10^−2^
	Rho-GDI signalling	-	2.14 × 10^−2^	1.00 × 10^−3^
	GABA receptor signalling	-	2.88 × 10^−2^	3.89 × 10^−2^
	Xenobiotic metabolism signalling	-	3.09 × 10^−2^	2.51 × 10^−3^
	EIF2 signalling	-	3.89 × 10^−2^	5.75 × 10^−2^
	Actin cytoskeleton signalling	-	4.07 × 10^−2^	8.71 × 10^−3^
	FAK signalling	-	4.57 × 10^−2^	1.66 × 10^−2^
	Integrin signalling	-	8.13 × 10^−2^	5.01 × 10^−2^
	RAC signalling	-	9.55 × 10^−2^	4.37 × 10^−2^

**Table 3 cells-10-02162-t003:** Summary of significant hits for related diseases identified using IPA Core Analysis. The disorders are significantly altered due to tau overexpression, to LMTM treatment or both. Significant alterations were found by linking our proteomics data with the Ingenuity Knowledge Base. The diseases linked to protein changes are named and corresponding *p*-values are given.

		*p*-Value		
	DISEASES	L66 vs.	L66 (15mg/kg MT)	L66 (45mg/kg MT)
		Wild-Type Controls	vs. L66 (Vehicle Control)	vs. L66 (Vehicle Control)
Genotype	Memory	3.63 × 10^−3^	-	-
	Behaviour	2.95 × 10^−2^	-	-
Genotype and LMTM	Alzheimer’s disease	3.31 × 10^−4^	3.39 × 10^−3^	2.24 × 10^−3^
	Degeneration of brain	5.50 × 10^−3^	5.89 × 10^−4^	1.20 × 10^−3^
LMTM	Disorder of basal ganglia	-	2.63 × 10^−8^	2.00 × 10^−7^
	Progressive motor neuropathy	-	9.12 × 10^−8^	5.37 × 10^−8^
	Neuromuscular disease	-	4.37 × 10^−7^	2.63 × 10^−6^
	Movement disorders	-	6.17 × 10^−7^	1.10 × 10^−6^
	Parkinson’s disease	-	8.71 × 10^−7^	2.29 × 10^−6^
	Neurodegeneration of hippocampus	-	9.12 × 10^−3^	1.10 × 10^−2^
	Cognitive impairment	-	2.75 × 10^−2^	2.69 × 10^−3^

**Table 4 cells-10-02162-t004:** Summary of putative upstream regulators identified using IPA Core Analysis. These regulatory molecules are significantly altered and upstream of pathways stated in Table 2.

	UPSTREAM	L66 vs.		L66 (15mg/kg MT)		L66 (45mg/kg MT)	
	REGULATORS	Wild-Type Controls		vs. L66 (Vehicle Control)		vs. L66 (Vehicle Control)	
		*p*-Value	Activation z-Score	*p*-Value	Activation z-Score	*p*-Value	Activation z-Score
Genotype	PLCB4	5.89 × 10^−3^	-	-	-	-	-
	Sept5	1.17 × 10^−2^	-	-	-	-	-
	ZNF746	1.74 × 10^−2^	-	-	-	-	-
	AKT1	2.34 × 10^−2^	-	-	-	-	-
	SLC18A3	2.34 × 10^−2^	-	-	-	-	-
	AGRP	2.88 × 10^−2^	-	-	-	-	-
	CDK5	3.47 × 10^−2^	-	-	-	-	-
	CDK5R1	3.47 × 10^−2^	-	-	-	-	-
	NOS2	3.98 × 10^−2^	-	-	-	-	-
	MAPT	1.35 × 10^−14^	-	4.90 × 10^−38^	-	2.75 × 10^−37^	-
Genotype and LMTM	PSEN1	1.05 × 10^−11^	-	9.12 × 10^−32^	-	5.75 × 10^−33^	-
	APP	6.31 × 10^−11^	-	2.14 × 10^−31^	-	1.45 × 10^−30^	-
	HTT	1.41 × 10^−7^	-	1.10 × 10^−9^	-	3.89 × 10^−8^	-
	SNCA	2.95 × 10^−4^	-	9.12 × 10^−10^	-	2.63 × 10^−9^	-
	YWHAG	5.01 × 10^−4^	-	4.57 × 10^−3^	-	5.50 × 10^−3^	-
	FMR1	1.51 × 10^−3^	-	3.39 × 10^−6^	-	6.61 × 10^−6^	-
	RTN4	1.91 × 10^−3^	-	6.46 × 10^−3^	-	9.12 × 10^−3^	-
	MKNK1	1.48 × 10^−2^	-	3.16 × 10^−6^	-	7.76 × 10^−6^	-
	HSF1	3.47 × 10^−2^	-	4.57 × 10^−3^	-	5.50 × 10^−3^	-
	BDNF	3.89 × 10^−2^	−1.12	1.66 × 10^−4^	−0.92	4.37 × 10^−4^	−0.92
LMTM	MTOR	-	-	2.04 × 10^−8^	2.1 (active)	5.01 × 10^−9^	1.7 (active)
	ADORA2A	-	-	3.63 × 10^−8^	−0.58	1.17 × 10^−9^	−1.1
	SOD1	-	-	2.04 × 10^−5^	-	2.45 × 10^−4^	-
	E2F1	-	-	2.24 × 10^−3^	-	3.47 × 10^−3^	-
	PARK2	-	-	1.05 × 10^−2^	-	1.26 × 10^−2^	-
	ARX	-	-	1.78 × 10^−2^	-	1.95 × 10^−2^	-
	EIF2B2	-	-	1.78 × 10^−2^	-	1.95 × 10^−2^	-
	MARK2	-	-	1.78 × 10^−2^	-	1.95 × 10^−2^	-
	MYO5A	-	-	1.78 × 10^−2^	-	1.95 × 10^−2^	-

## Data Availability

All data are provided within the manuscript of the Appendix A.

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
