# Peer review of "Proteomic Analysis of Hydromethylthionine in the Line 66 Model of Frontotemporal Dementia Demonstrates Actions on Tau-Dependent and Tau-Independent Networks"

_cells, 2021, doi:10.3390/cells10082162_

Round 1
Reviewer 1 Report
The manuscript is well written, and the proteomic findings are interesting. However, there are a great deal of loose endings in their work and a massive part of the differences in protein level identified is not validated which posses a problem with the certainty of the post-hoc analyses. As such, although the authors provide western blot data in support of the impact of LMM on synaptic and neuronal function, they do not show data that supports the effect of LMTM on NRF2-mediated oxidative stress response and on mitochondria neither on the impact of LMTM on protein degradation. A few measures or experiments supporting the 2 pathways are needed.
The manuscript also contains several overclaims. One example is that the authors state that the decrease in BDNF might put synapses at risk but then count NeuN immunoreactive neurons. As there is a difference between synaptic pathology and neuronal cell death, the authors should have examined a synaptic marker instead. Another example is that the authors say: “It is therefore not surprising that mitochondrial function was altered in L66 and recovered following LMTM treatment (Table 2)”. However, the authors do not show any experiments related to mitochondrial function. Therefore, lines 522 to 532 should be rephrased to reflect the fact that their statements are based on proteomic observations alone.
Other minor comments:
It is unfortunate that the proteomic analysis does not include a group of wild-type LMTM treated mice. It would allow to best delineate “off-target” effects of LMTM.
The authors should better define the reasons for the choice of a 2DE analysis combined with nanoLC-ESI-MS/MS of excised spots. 2DE often underestimates proteomic changes and more advanced quantitative proteomic techniques are now available.
For those who are not familiar with this line, it should be explained why the experiments focused on females.
Author Response
Reviewer 1:
- The manuscript is well written, and the proteomic findings are interesting. However, there are a great deal of loose endings in their work and a massive part of the differences in protein level identified is not validated which possess a problem with the certainty of the post-hoc analyses. As such, although the authors provide western blot data in support of the impact of LMM on synaptic and neuronal function, they do not show data that supports the effect of LMTM on NRF2-mediated oxidative stress response and on mitochondria neither on the impact of LMTM on protein degradation. A few measures or experiments supporting the 2 pathways are needed.
Reply: We thank the reviewer for the careful analysis of our work. Although we do not show validation of the 2D mass spec results for the above-mentioned pathways with another method, e.g. western blots, we consider that the results would be valid. We have indeed placed synaptic and neuronal function pathways at the heart of the manuscript and these were chosen for detailed validation. Western blots confirmed the 2D MS results and we therefore feel it opportune to also discuss other pathways in which we have shown alterations based solely on the proteomic analysis. 2DE mass spec approaches (and subsequent pathway analyses) typically result in large data sets and reveal numerous affected pathways. A complete validation is highly unusual and certainly beyond the scope of our study as reported here.
- The manuscript also contains several overclaims. One example is that the authors state that the decrease in BDNF might put synapses at risk but then count NeuN immunoreactive neurons. As there is a difference between synaptic pathology and neuronal cell death, the authors should have examined a synaptic marker instead. Another example is that the authors say: “It is therefore not surprising that mitochondrial function was altered in L66 and recovered following LMTM treatment (Table 2)”. However, the authors do not show any experiments related to mitochondrial function. Therefore, lines 522 to 532 should be rephrased to reflect the fact that their statements are based on proteomic observations alone.
Reply: The use of a synaptic marker for synapse quantification is certainly a very good suggestion and we will certainly take this on board for upcoming experiments
As for BDNF, we would like to highlight that this molecule is not only implicated in synaptic but also in neuronal functions. We now have included this fact in the amended highlighted text (page 13). Since we discuss the implication of BDNF on neurones in neurodegenerative diseases (page 18), we felt it more appropriate to quantify neurones rather than synapses.
Although not presented in detail, the entire IPA analysis is based on our 2D mass spec and downstream pathway analyses. This uncovered alterations in pathways including those involved in mitochondrial function. So the reviewer is not correct in her/his suggestion that no results are presented from such an analysis. She/he is correct, however that we do not show additional experiments to validate this function, but this point has been discussed above under #1.
Other minor comments:
- It is unfortunate that the proteomic analysis does not include a group of wild-type LMTM treated mice. It would allow to best delineate “off-target” effects of LMTM.
Reply: We fully agree with this reviewer and would include such cohorts in future experiments. Unfortunately, this would take considerable effort and time.
- The authors should better define the reasons for the choice of a 2DE analysis combined with nanoLC-ESI-MS/MS of excised spots. 2DE often underestimates proteomic changes and more advanced quantitative proteomic techniques are now available.
Reply: The reviewer is indeed correct, there are other quantitative proteomics techniques now available. However, we favoured the gel-based method because of its potential in examining proteins at the species level, including post-translational modifications. We feel that both bottom-up and top-down approaches have their merits and our expertise with the 2DE approach was therefore preferred.
- For those who are not familiar with this line, it should be explained why the experiments focused on females.
Reply: These mice have been characterised in a number of publications from our group. However, we have extended a brief description of the mice in the Method section – animal part (page 5).
Reviewer 2 Report
The introduction section is unconventional. It read like a review article. It is necessary to rewrite it in a more precise manner focusing on the essential information for the study, in particular why carry out proteomics analysis on an FTD model, what is this L66 line, and the rational of using LMTM to rescue the deficit. FTD are neurodegenerative diseases that affect mainly frontotemporal region. How well the L66 line mimics FTD (or tauopathy) in human? Is the whole brain affected by the overexpression, or specific brain regions are differentially affected. What is the rational of taking the whole brain for the analysis.
There are many recent proteomics studies on neurodegeneration. The review articles that the authors cited are not up-to-date (ref 11-16, published between 2008-2015). More recent reviews from the past few years should be cited instead.
In the abstract section the goal of the experiment is clearly stated, (1) reveal alteration of protein abundance by overexpression of Tau, (2) proteins/pathways that are rescued by LMTM administration, and (3) the LMTM-induced changes independent of Tau overexpression. However, in the main text, the description of the results are difficult to follow.
In experiment 1. the 2D gel revealed 71 spots (=123 protein) that have differential abundance between wt and L66 mice. If LMTM administration has recue effect, it would be informative to show the changes of fold differences between L66 and L66+LMTM groups on these 71 spots (not only the 17 common spots). It is possible that the spot may show partial rescue, although it may not reach significant level. One can hypothesize that proteins contained in the same organelles/molecular pathways would be co-regulated in the same direction. The analysis of these 72 spots (123 proteins) may reveal their co-regulation, and that in turn may implicate the regulation of specific biological processes. For example, are mitochondrial proteins globally regulated in the same direction?
In experiment 2, I suppose the fold changes (of the spots) can be expressed as their ratios between L66 and the L66+LMTM groups. The authors stated that : “The majority of these proteins (about 80%) showed high abundance in the vehicle cohort (see green cluster in the upper left quadrant of the heatmap Fig. 4C), most of which were downregulated by both doses of LMTM. About half of these proteins were lowered to the average expression level (black); the other half was decreased below average expression (red). Other proteins (about 20%) were at a low level of expression for vehicle-treated mice”. This paragraph is very difficult to follow. Did the study reveal simply xx upregulated and yy downregulated proteins?
Line 393, “A greater resolution of these protein activities was gained through Ingenuity Path- way Analysis (IPA)”. It really depends on the number of proteins used for the analysis, and that the up- and down-regulated proteins must be analyzed separately. The redundant terms should be removed. For example, synaptic transmission of neurons, neurotransmission, synaptic depression, synaptic transmission are shown, but all these proteins point to the same process.
The authors performed western blotting to examine the changes of selected proteins in mutant mice and the treatment with LMTM. How many of them were selected based on the results of 2D gel experiments, or they were selected based on hypothesis? Could the observed changes of proteins from the 2D gel experiments be validated by western blotting analysis?
In the discussion section, the authors stated that “the changes are most likely associated with neurons, since these cells account for around two-third of the different cell types in the mouse brain [66] and tau aggregates are most prevalent in neurons”. Why? As far as I know, astrocyte and microglia are highly activated in brain regions that suffer from neurodegeneration.
Specific questions
Line 246: “Raw spectra files were generated by DataAnalysis 3.2 (Bruker, Germany), uploaded to the MaxQuant platform (version 1.5.1.2, Max Planck Institute, Germany) and searched 247 against a UniProtKB mouse protein data base”.
Maxquant accepts .Raw file from Thermo. It is not clear to me why Bruker DataAnalysis was used to generated raw spectra files.
Lin 250: “The MaxQuant software allows protein identification and label-free quantification (LFQ; based on peak intensities of spectra)”.
The authors correctly pointed out that label-free quantification is a useful method for quantitative proteomics. Therefore, I find it strange that they did not perform proteomics analysis by LC-MS/MS. The authors should at least mention the limitation of 2D gel (that membrane receptors etc are not detected, single spot may contain multiple proteins, single protein may be present in multiple spots), and discuss the use of DDA/DIA for future studies.
Author Response
Reviewer 2:
- The introduction section is unconventional. It read like a review article. It is necessary to rewrite it in a more precise manner focusing on the essential information for the study, in particular why carry out proteomics analysis on an FTD model, what is this L66 line, and the rational of using LMTM to rescue the deficit. FTD are neurodegenerative diseases that affect mainly frontotemporal region. How well the L66 line mimics FTD (or tauopathy) in human? Is the whole brain affected by the overexpression, or specific brain regions are differentially affected. What is the rational of taking the whole brain for the analysis.
Reply: The focus and rationale of this work are clearly stated at the end of the first paragraph of the Introduction. ‘ This work constitutes one of the first attempts to determine such global alterations in brain protein expression as a result of drug treatment.’ We felt that it was necessary, therefore. to lay out the background of the methodology, its use in human and animal samples and, given we report the first proteomic analysis using drug interference in an FTD model, some information on the rationale for treatment regime and therapeutic (see first sentence of last paragraph of Introduction: ‘Prevention of tau aggregation is a valid therapeutic strategy since numerous studies have confirmed a correlation between the degree of tau aggregation and the severity of dementia [12,13,35,43].‘ This was contained within 2.5 pages. Moreover, we find it somehow inconsistent that the reviewer requested further literature to be included in the Introduction (see point 2).
As for a more detailed description of Line 66 mice, we have added some information in the Method section on Animals (page 5) to highlight our current knowledge of this model.
- There are many recent proteomics studies on neurodegeneration. The review articles that the authors cited are not up-to-date (ref 11-16, published between 2008-2015). More recent reviews from the past few years should be cited instead.
Reply: More recent citations, including reviews and empirical articles have been added to the introduction now and the numbering of citations was amended. We should note that all of these citations are related to Alzheimer’s disease, and not FTD.
These citations were added:
Jain, A.P.; Sathe, G. Proteomics landscape of Alzheimer’s disease. Proteomes 2021, 9, 1–18, doi:10.3390/proteomes9010013.
Tanaka, T.; Lavery, R.; Varma, V.; Fantoni, G.; Colpo, M.; Thambisetty, M.; Candia, J.; Resnick, S.M.; Bennett, D.A.; Biancotto, A.; et al. Plasma proteomic signatures predict dementia and cognitive impairment. Alzheimer’s Dement. Transl. Res. Clin. Interv. 2020, 6, 1–11, doi:10.1002/trc2.12018.
Bader, J.M.; Geyer, P.E.; Müller, J.B.; Strauss, M.T.; Koch, M.; Leypoldt, F.; Koertvelyessy, P.; Bittner, D.; Schipke, C.G.; Incesoy, E.I.; et al. Proteome profiling in cerebrospinal fluid reveals novel biomarkers of Alzheimer’s disease. Mol. Syst. Biol. 2020, 16, 1–17, doi:10.15252/msb.20199356.
Whelan, C.D.; Mattsson, N.; Nagle, M.W.; Vijayaraghavan, S.; Hyde, C.; Janelidze, S.; Stomrud, E.; Lee, J.; Fitz, L.; Samad, T.A.; et al. Multiplex proteomics identifies novel CSF and plasma biomarkers of early Alzheimer’s disease. Acta Neuropathol. Commun. 2019, 7, 1–14, doi:10.1186/s40478-019-0795-2.
- In the abstract section the goal of the experiment is clearly stated, (1) reveal alteration of protein abundance by overexpression of Tau, (2) proteins/pathways that are rescued by LMTM administration, and (3) the LMTM-induced changes independent of Tau overexpression. However, in the main text, the description of the results are difficult to follow.
Reply: As requested by this reviewer, we have now re-arranged and re-named some of the subsections in the Results. The manuscript now follows our goals more directly and is sectioned into: (1) overview section about number of differentially abundant proteins, (2) alteration of protein abundance induced by overexpression of Tau, (3) alteration of protein abundance induced by LMTM, (4) tau-dependent pathway rescue by LMTM and (5) tau-independent pathway rescue by LMTM. The sequence of some subsections and their headings (not their text) has been rearranged as a consequence.
We trust this enables readers of our paper to follow the Results more readily.
- In experiment 1. the 2D gel revealed 71 spots (=123 protein) that have differential abundance between wt and L66 mice. If LMTM administration has recue effect, it would be informative to show the changes of fold differences between L66 and L66+LMTM groups on these 71 spots (not only the 17 common spots). It is possible that the spot may show partial rescue, although it may not reach significant level. One can hypothesize that proteins contained in the same organelles/molecular pathways would be co-regulated in the same direction. The analysis of these 72 spots (123 proteins) may reveal their co-regulation, and that in turn may implicate the regulation of specific biological processes. For example, are mitochondrial proteins globally regulated in the same direction?
Reply: An analysis as suggested by the reviewer is theoretically possible, but only under the premise that ALL molecules in a given pathway are regulated in the same direction (up or down). However, most pathways contain counter-regulated proteins; that means that while one molecule goes up, the others go down and decrease inhibition to the third one. Therefore, a significant and meaningful change in a given pathway (for example mitochondrial proteins) is only possible if our cut-off points are implemented. These limits are clearly stated in the Method section (“Spots were considered to be regulated when the relative spot density was significantly different (p < 0.05) and if the spot was present in at least 80% of gels in each study group. No further cut-off or post-hoc analyses were applied to the data and all significantly regulated spots with the above-described behaviour were subjected to MS analysis.”). These limits were set a priori of any further analyses.
- In experiment 2, I suppose the fold changes (of the spots) can be expressed as their ratios between L66 and the L66+LMTM groups. The authors stated that : “The majority of these proteins (about 80%) showed high abundance in the vehicle cohort (see green cluster in the upper left quadrant of the heatmap Fig. 4C), most of which were downregulated by both doses of LMTM. About half of these proteins were lowered to the average expression level (black); the other half was decreased below average expression (red). Other proteins (about 20%) were at a low level of expression for vehicle-treated mice”. This paragraph is very difficult to follow. Did the study reveal simply xx upregulated and yy downregulated proteins?
Reply: We have now added numbers of proteins corresponding to the percentage and highlighted these in the text (page 11, penultimate paragraph).
- Line 393, “A greater resolution of these protein activities was gained through Ingenuity Path- way Analysis (IPA)”. It really depends on the number of proteins used for the analysis, and that the up- and down-regulated proteins must be analyzed separately.
Reply: The first part of the reviewer’s statement is an inherent element of any proteome analysis. The second part of the statement, however, is questionable. (i) Firstly, it directly contradicts point 4 of the reviewer’s comments suggesting an analysis of co-regulated proteins pertaining to the same pathway; (ii) Secondly, what is critical is the effect that a drug has on a particular pathway and subsequent changes, rather than on a single step within that pathway. For some pathways, all proteins may be positively correlating (e.g. either all up-regulated or all down-regulated); for other pathways, certain proteins can be negatively correlating (e.g. protein 1 is increased and that causes a decrease in proteins 2 and 3, which may result, for example, in an overall activation of the pathway). Thus, analysing up- and down-regulated proteins separately is not a recommended approach in our opinion.
- The redundant terms should be removed. For example, synaptic transmission of neurons, neurotransmission, synaptic depression, synaptic transmission are shown, but all these proteins point to the same process.
Reply: We fully agree with the logic in the argument of the reviewer. However, the specific terms in the text stem directly from the IPA output and we are reluctant to alter these terms and thereby modify any precision placed in them by the software.
- The authors performed western blotting to examine the changes of selected proteins in mutant mice and the treatment with LMTM. How many of them were selected based on the results of 2D gel experiments, or they were selected based on hypothesis? Could the observed changes of proteins from the 2D gel experiments be validated by western blotting analysis?
Reply: In principle, proteomics is a hypothesis-free approach, and we have added this now in the first paragraph in the Results (page 10). This means that no proteins were selected a priori, they were only selected based on IPA output. Based on 600+ changes in protein species abundance, we first performed a detailed pathway analyses using IPA. The software predicts the most prominent pathways affected and we have selected some of the key proteins for validation by western blot. The selection criteria are stated in the text (for example “Accordingly, among the 27 protein species dysregulated by tau-overexpression and modulated by LMTM, several protein species have known functions in synaptic/neurotransmission” and “PI3K/AKT involved in signal transduction and the mTOR kinase P70S6K were significantly altered in line with the IPA analyses (Table 2)”).
- In the discussion section, the authors stated that “the changes are most likely associated with neurons, since these cells account for around two-third of the different cell types in the mouse brain [66] and tau aggregates are most prevalent in neurons”. Why? As far as I know, astrocyte and microglia are highly activated in brain regions that suffer from neurodegeneration.
Reply: The reviewer is indeed correct: astrocytes and microglia are highly activated during neurodegenerative events, and we do not dispute this, but how they are activated has not been determined (Bussian et al., Nature 2018, Clearance of senescent glial cells prevents tau-dependent pathology and cognitive decline). Our wording was carefully selected to account for this unknown effect, and the facts that (i) the use of the Thy1 promotor in our transgenic mice induces neurone-specific expression of full-length human tau, and (ii) that FTD-tau cases show preferential aggregation of tau in specific neuronal cell types (Lin et al., Acta Neuropathol Commun 2019, Preferential tau aggregation in von Economo neurons and fork cells in frontotemporal lobar degeneration with specific MAPT variants). We have added a sentence to reinforce this statement (page 15, first paragraph of Discussion).
Specific questions
- Line 246: “Raw spectra files were generated by DataAnalysis 3.2 (Bruker, Germany), uploaded to the MaxQuant platform (version 1.5.1.2, Max Planck Institute, Germany) and searched 247 against a UniProtKB mouse protein data base”. Maxquant accepts .Raw file from Thermo. It is not clear to me why Bruker DataAnalysis was used to generated raw spectra files.
Reply: We thank the reviewer for bringing this mistake to our attention. The software used was indeed Xcalibur 2.1 from Thermo Fisher Scientific. This has now been amended in the text (page 8).
- Line 250: “The MaxQuant software allows protein identification and label-free quantification (LFQ; based on peak intensities of spectra)”.
The authors correctly pointed out that label-free quantification is a useful method for quantitative proteomics. Therefore, I find it strange that they did not perform proteomics analysis by LC-MS/MS. The authors should at least mention the limitation of 2D gel (that membrane receptors etc are not detected, single spot may contain multiple proteins, single protein may be present in multiple spots), and discuss the use of DDA/DIA for future studies.
Reply: In this context, the reviewer may also find our comment to point 4 of Reviewer 1 instructive. Indeed, detection of multiple proteins in a single spot or a single protein in multiple spots are genuine advantages of our gel-based approach and enable examination of proteins at the species level. In our experience, this approach also allows the detection of membrane receptors because we used a urea-based protein extraction method, and indeed we were able to detect membrane receptors and transmembrane channels in the current study (e.g. Gm7120, Tmem185a, Vstm2a, Vdac, Kcnab2). In contrast to the reviewer’s suggestion, we would argue that membrane receptors are less likely to be detected by LC-MS because they are retained in the HPLC columns due to their hydrophobicity and may thus be depleted in the analyte that flows to and gets detected by the mass spec.
Reviewer 3 Report
The manuscript entitled “Proteomic analysis of hydromethylthionine in the line 66 model of frontotemporal dementia demonstrates actions on tau-dependent and tau-independent networks” is a well-written and detailed manuscript. In this manuscript, the authors revealed how LMTM, a tau aggregation inhibitor, changes the protein network. LMTM altered both of tau-dependent and independent pathways, which are interesting findings of this manuscript. However, there are some concerns that should be addressed before publication.
Major issues
1: To link their findings on proteome and pathology, they should confirm some key protein changes in the brain regions they could find the genotype effects.
2: They also should discuss why LMTM could not reverse the pathological changes induced by the mutant tau expression in some brain regions.
Minor issues
1: The introduction included non-necessary descriptions that are more appropriate for discussion section.
2: It is unclear why the authors colored some sentences in red and crossed out with lines. The manuscript appears to be a draft version.
3: Tables appear to be cropped partially. Please carefully check the appearance of the whole manuscript again.
4: The authors used the rare spell, “neurones” in the text and “neurons” in tables. It will be better to used “neurons” to unify the spell throughout the manuscript.
5: In line 691, there is a typo ‘dysfunction. synaptic transmission and stress responses.’. The sentence should be “dysfunction, synaptic transmission, and stress responses.”
Author Response
Reviewer 3
Major issues
- To link their findings on proteome and pathology, they should confirm some key protein changes in the brain regions they could find the genotype effects.
Response: The proteomics analysis (or western blot confirmation of protein levels) was performed on whole brains (without olfactory bulbs), as stated in the Method section - Sample collection part (page 6). As the tau transgene in L66 is under control of the neuron-specific Thy-1 regulatory element (previously described in detail in Melis et al, Cell. Mol. Life Sci. 2015 and Melis et al, Behav. Pharmacol. 2015), tau pathology is predominant in iso- and allocortex, but also found globally in motor and basal forebrain regions. While we agree in principle to the reviewer’s suggestion that a richer understanding would be achievable through proteomic analyses of specific brain regions, we have no evidence that expression levels differ dramatically between these areas. Furthermore, we had indicated this specific limitation at the beginning of the discussion “A limitation of this study, however, is that the resulting proteomic analyses do not provide spatial resolution of FTD-related brain areas but rather reflect changes that occur across the whole brain”. Our focus was on the drug-dependent changes rather than region-specificity.
- They also should discuss why LMTM could not reverse the pathological changes induced by the mutant tau expression in some brain regions.
Response: As highlighted under query 1 of this reviewer, we did not perform regional analyses and our promotor used for transgene expression does not discriminate brain regions. Therefore, it is impossible to differentiate effects of LMTM on brain structures (as already pointed out in Melis et al, Behav. Pharmacol. 2015). What we have discussed, however, is the correction by LMTM of certain global protein changes, in the absence of others (such as BDNF and its downstream targets) (page 17) and also made a case for effective drug doses that were administered. Again, we did not conduct region-specific proteomics and have never reported region-specific changes in pathology under LMTM.
Minor issues
- The introduction included non-necessary descriptions that are more appropriate for discussion section.
Response: This work constitutes one of the first and few attempts to determine global alterations in brain protein expression as a result of therapeutically relevant drug treatment. We feel strongly that both the background of the methodology, the use of the genetically altered model and the drug approach need explanation. All this information was compressed to 2.5 pages of Introduction, which we believe is not excessive.
- It is unclear why the authors colored some sentences in red and crossed out with lines. The manuscript appears to be a draft version.
Response: This third reviewer has received a revised version (R1) of our manuscript and highlights and colourings were introduced to mark responses to reviewers 1 and 2 of our original submission. Apologies that this was not pointed out.
- Tables appear to be cropped partially. Please carefully check the appearance of the whole manuscript again.
Response: Many thanks for highlighting this issue. We have now carefully checked all tables (and each cell) and did not find any of them wrongly formatted. Tables 1-3 have 5 columns and Table 4 has 8 columns. All Tables appear to be correct in our text version (word document). We would like this reviewer to re-check and return to us should this issue be reappearing.
- The authors used the rare spell, “neurones” in the text and “neurons” in tables. It will be better to used “neurons” to unify the spell throughout the manuscript.
Response: Neurones has now been amended to neurons throughout the manuscript.
- In line 691, there is a typo ‘dysfunction. synaptic transmission and stress responses.’. The sentence should be “dysfunction, synaptic transmission, and stress responses.”
Response: Thanks. This has now been corrected.
Round 2
Reviewer 1 Report
I would like to thank the authors for their answers. Although this reviewer agrees that adding additional experiments would be a lengthy process, some minor text revisions would add clarity. This reviewer suggests to replace sentence “It is therefore not surprising that mitochondrial function was altered in L66 and recovered following LMTM treatment (Table 2)” by a milder statement such as "“It is therefore not surprising that the expression of proteins related to mitochondrial function was altered in L66 and recovered following LMTM treatment (Table 2)”. Similarly, sentence "Since it is conceivable that the lack of trophic and structural support via BDNF and -synuclein puts synapses at risk, we next sought to confirm this by quantitative counting of NeuN immunoreactive neurones in tau-rich cortical areas comparing genotypes and drug cohorts" should be modified as previously requested to better reflect that NeuN is not indicative of synapse health.
Author Response
Reviewer 1
I would like to thank the authors for their answers. Although this reviewer agrees that adding additional experiments would be a lengthy process, some minor text revisions would add clarity.
- This reviewer suggests to replace sentence “It is therefore not surprising that mitochondrial function was altered in L66 and recovered following LMTM treatment (Table 2)” by a milder statement such as "“It is therefore not surprising that the expression of proteins related to mitochondrial function was altered in L66 and recovered following LMTM treatment (Table 2)”.
Response: The sentence has been amended as requested by the reviewer (Results part, page 14).
- Similarly, sentence "Since it is conceivable that the lack of trophic and structural support via BDNF and a-synuclein puts synapses at risk, we next sought to confirm this by quantitative counting of NeuN immunoreactive neurones in tau-rich cortical areas comparing genotypes and drug cohorts" should be modified as previously requested to better reflect that NeuN is not indicative of synapse health.
Response: The sentence has been amended as requested by the reviewer (Results part, page 13).
Reviewer 2 Report
There is still not much improvement in this revised version.
The authors fail to address most of my suggestions that may help to improve the manuscript.
For example, I asked them to include recent publications on proteomics of FTD. Their reply "More recent citations, including reviews and empirical articles have been added to the introduction now and the numbering of citations was amended. We should note that all of these citations are related to Alzheimer’s disease, and not FTD". There are many FTD/ALS proteomics studies, including recent review articles (e.g. Front. Neurosci., 11 June 2019 | https://doi.org/10.3389/fnins.2019.00548. Proteomics Approaches for Biomarker and Drug Target Discovery in ALS and FTD). I do not understand why they refused to cite the relevant articles.
The authors used 2D gel technology for the analysis. This method has many limitation, and the data are generally very difficult to interpret. It is largely abandoned by the proteomics community. Nevertheless, I tried to give hints so that they may improve the manuscript to a level that may be publishable. They simply ignored the advices.
Another example, they stated that "In contrast to the reviewer’s suggestion, we would argue that membrane receptors are less likely to be detected by LC-MS because they are retained in the HPLC columns due to their hydrophobicity and may thus be depleted in the analyte that flows to and gets detected by the mass spec". I wonder if they even know how bottom-up proteomics works, because what they stated is largely wrong.
To me, the manuscript remains unpublishable.
Author Response
There is still not much improvement in this revised version. The authors fail to address most of my suggestions that may help to improve the manuscript.
Response: It is disappointing to us that this reviewer is still unsatisfied with our replies. It is indeed the case that in recent years, more proteomic studies have used LC-MS/MS technologies, but this should not be the reason for being so dismissive of the 2DE approach. It is therefore clear to us, that we may never be able address all issues criticized by this reviewer. We have nevertheless tried to include limitations of the method in the main text (including citations that have compared the methodologies), but this does not mean our method and data are invalid. To the contrary, the results largely conform and expand recent work using LC-MS/MS technologies underlining the validity of our approach.
- For example, I asked them to include recent publications on proteomics of FTD. Their reply "More recent citations, including reviews and empirical articles have been added to the introduction now and the numbering of citations was amended. We should note that all of these citations are related to Alzheimer’s disease, and not FTD". There are many FTD/ALS proteomics studies, including recent review articles (e.g. Front. Neurosci., 11 June 2019 | https://doi.org/10.3389/fnins.2019.00548. Proteomics Approaches for Biomarker and Drug Target Discovery in ALS and FTD). I do not understand why they refused to cite the relevant articles.
Response: This was an oversight on our part, for which we apologise. We have now included the Frontiers in Neuroscience publication in the main text and mention others (Iridoy et al., 2918 and Swift et al. 2021) which are more empirical studies or report on a fluidity analysis. We should note that our search did not include ALS as a search term so ‘many publications were specifically excluded from our manuscript and deemed irrelevant.
- The authors used 2D gel technology for the analysis. This method has many limitation, and the data are generally very difficult to interpret. It is largely abandoned by the proteomics community. Nevertheless, I tried to give hints so that they may improve the manuscript to a level that may be publishable. They simply ignored the advices.
Response: In our previous response, this has been addressed, in our response to point 1. The critique was mainly centred around the animal model used and the reviewer asked for information on the suitability of this model as an FTD-mimicking experimental tool. This was not ignored. Information and details on the expression pattern in the transgenic mouse model is provided in the text (see Methods, animals) and through cited literature. As for the limitations of our 2DE method, we do not dispute those. These limitations, however, do not invalidate the data and we have been cautious with our interpretation of the data by taking these limitations on board. However, there are clear advantages to our methodology as the quantification of proteins at the species level is fundamental to determine their function in disease and can only be achieved by top-down approaches with flexible and tunable resolution (Jungblut et al, The speciation of the proteome, Chem Cent J. 2008; Ning et al, Exploiting the potential of 2DE in proteomics analyses, Expert Rev Proteomics. 2016; Zhan et al, Revival of 2DE-LC/MS in Proteomics and Its Potential for Large-Scale Study of Human Proteoforms, Med One. 2018; Zhan et al, Innovating the Concept and Practice of Two-Dimensional Gel Electrophoresis in the Analysis of Proteomes at the Proteoform Level, Proteomes. 2019). This was also added in the Discussion (page 15, paragraph1).
- Another example, they stated that "In contrast to the reviewer’s suggestion, we would argue that membrane receptors are less likely to be detected by LC-MS because they are retained in the HPLC columns due to their hydrophobicity and may thus be depleted in the analyte that flows to and gets detected by the mass spec". I wonder if they even know how bottom-up proteomics works, because what they stated is largely wrong. To me, the manuscript remains unpublishable.
Response: It appears that there is a contrast of opinion. Therefore, we have consulted with chemists with years of expertise in HPLC and they have ‘largely’ confirmed our statement. We respect the reviewer’s point of view and feel that we cannot address this comment any further.
Reviewer 3 Report
All of the points I raised have been addressed.Author Response
We thanks this reviewer for her/his comments.
Round 3
Reviewer 2 Report
The Orbitrap was operated in data-dependent mode by subjecting the ten most abundant ions of each survey spectrum (nominal resolution 35,000) to higher-energy C-trap dissociation-fragmentation.
The phase “higher-energy C-trap dissociation-fragmentation “ is confusing. I think the Orbitrap XL was operated as follow: “The precursor ion was sent into gas filled collision cell for high energy collisional dissociation. The fragment ions were sent to C-trap, and transferred to orbitrap for analysis”.
Author Response
Cells 2021: proteomics in L66 – R3 -specific comments
Reviewer 2
The Orbitrap was operated in data-dependent mode by subjecting the ten most abundant ions of each survey spectrum (nominal resolution 35,000) to higher-energy C-trap dissociation-fragmentation.
The phase “higher-energy C-trap dissociation-fragmentation“ is confusing. I think the Orbitrap XL was operated as follow: “The precursor ion was sent into gas filled collision cell for high energy collisional dissociation. The fragment ions were sent to C-trap and transferred to orbitrap for analysis”.
Response: We thank the reviewer for bringing this to our attention and we have now corrected the sentence as requested.